

# An Evaluation of Biomass Burning Aerosol Mass, Extinction, and Size Distribution in GEOS using Observations from CAMP²Ex

Allison B. Marquardt Collow[1,2], Virginie Buchard[1,2], Peter R. Colarco[2], Arlindo M. da Silva[2], Ravi Govindaraju[2,3], Edward P. Nowottnick[2], Sharon Burton[4], Rich Ferrare[4], Chris Hostetler[4], and Luke Ziemba[4]

[1]University of Maryland Baltimore County, Baltimore, Maryland, USA
[2]NASA Goddard Space Flight Center, Greenbelt, Maryland, USA
[3]Science Systems and Applications, Inc., Lanham, Maryland, USA
[4]NASA Langley Research Center, Hampton, Virginia, USA

*Correspondence to*: Allison Collow (allison.collow@nasa.gov)

**Abstract.** Biomass burning aerosol impacts aspects of the atmosphere and Earth system through radiative forcing, serving as cloud condensation nuclei, and air quality. Despite its importance, the representation of biomass burning aerosol is not always accurate in numerical weather prediction and climate models or reanalysis products. Using observations collected as part of the Cloud, Aerosol and Monsoon Processes Philippines Experiment (CAMP²Ex) in August through October of 2019, aerosol concentration and optical properties are evaluated within the Goddard Earth Observing System (GEOS) and its underlying aerosol module, GOCART. In the operational configuration, GEOS assimilates aerosol optical depth observations at 550 nm to constrain aerosol fields. Particularly for biomass burning aerosol, without the assimilation of aerosol optical depth, aerosol extinction is underestimated compared to observations collected in the Philippines region during the CAMP²Ex campaign. The assimilation process adds excessive amounts of carbon to account for the underestimated extinction, resulting in positive biases in the mass of black and organic carbon, especially within the boundary layer, relative to in situ observations from the Langley Aerosol Research Group Experiment. Counteracting this, GEOS is deficient in sulphate and nitrate aerosol just above the boundary layer. Aside from aerosol mass, extinction within GEOS is a function of ambient relative humidity and an assumed particle size distribution. The relationship between dry and ambient extinction in GEOS reveals that hygroscopic growth is too aggressive within the model for biomass burning aerosol. An additional concern lies in the assumed particle size distribution for GEOS, which has a mode radius that is too small for organic carbon. Variability in the observed particle size distribution for biomass burning aerosol within a single flight also illuminates the fact that a single assumed particle size distribution is not sufficient and that for a proper representation, a more advanced aerosol module with GEOS may be necessary.

## 1 Introduction

Aerosols are an important component of the Earth system due to their role as cloud condensation nuclei, in extinction of radiation, and impact on air quality. It is therefore essential to be able to accurately capture their optical properties, transport, and overall life cycle in Earth system models. Field campaigns provide valuable data that can be used to evaluate models. One





such field campaign, the Cloud, Aerosol and Monsoon Processes Philippines Experiment (CAMP²Ex) in 2019, was based out
of the Philippines and had the opportunistic timing of being able to collect observations focused on the interaction of clouds,
aerosols, and radiation before, during, and after the transition of the Southwest Monsoon of the South China Sea. Boreal
summer in the Philippines region is characterized by winds out of the southwest (Wang et al., 2009) that transport smoke and
biomass burning aerosol into the Sulu and Philippine Seas (Xian et al., 2013). The maritime continent, particularly Borneo and
Sumatra, is susceptible to peatland fires during this time of year, which are exacerbated by drought and El Niño conditions
(Reid et al., 2012; Yin, 2020). Depending on the large-scale circulation, smoke can be the primary pollutant. Otherwise, the
relatively clean marine airmass can be polluted by plumes of urban aerosols, either locally from the Philippines or transported
from Asia. Heating due to biomass burning aerosol has been shown to feedback onto dynamics within the atmosphere, altering
vertical motion and therefore vertical profiles of gases such as water vapor and carbon monoxide (Ott et al., 2010). Aside from
the direct radiative effect on radiation (e.g., Chang et al., 2021), smoke can have a semi-direct effect in which the frequency
of clouds is altered (Mallet et al., 2020; Ding et al., 2021). Additionally, properties of biomass burning aerosols play a role on
cloud condensation nuclei concentration, their activation, and droplet formation (Chen et al., 2019; Li, 2019; Kacarab et al.,
2020; Zheng et al., 2020).

A common issue with climate models with respect to biomass burning aerosol is that it tends to be too absorbing
(Brown et al., 2021). However, variability exists in single scatter albedo (SSA) among models due to assumptions for aerosol
size distributions, mixing state, and refractive indices (Shinozuka et al., 2020; Brown et al., 2021). An additional source of
uncertainty arises with the emissions of biomass burning aerosols, which is responsible for inconsistent loading of organic
aerosol in models (Pan et al., 2020; Gliß et al, 2021). Emissions may even be tuned to achieve desirable values for total aerosol
optical depth (Petrenko et al., 2017). Varying complexities of parameterizations for secondary organic aerosol (SOA) and the
aging of organic aerosol result in a spread of organic aerosol loadings and lifetimes within models (Tsigaridis, 2014). Aging
characteristics of carbon have been found to be particularly important for modelling the direct aerosol forcing of black carbon
(Wang et al., 2018). Carbonaceous aerosols can have a wide array of characteristics depending on their source. Carbon emitted
through the combustion of wood can have a much lower water uptake than other fuels such as diesel (Wang et al., 2020) and
absorption properties are dependent on the chemical composition of the fuel type (Tang et al., 2020). It is therefore important
to distinguish between white (anthropogenetic) and brown (biomass burning) organic carbon and use appropriate optical
properties for each type of carbon.

One Earth system model that is subject to these uncertainties is the Goddard Earth Observing System (GEOS), with
the underlying Goddard Chemistry Aerosol Radiation and Transport (GOCART) model (Chin et al., 2002; Colarco et al.,
2010). A recent development aimed at improving the representation of biomass burning aerosol in GOCART is the introduction
of a brown carbon species (Colarco et al., 2017). Amid ongoing development in the physical parameterizations within GEOS,
it is important to evaluate changes that are made to GOCART as aerosols feed back into the Earth system through radiation.
Although two moment cloud microphysics is not used in operational version of GEOS at this time, providing accurate
representations of aerosol mass is a necessity for experimental GEOS simulations with two moment microphysics (Barahona



et al., 2014). Using the wealth of observational data collected during CAMP²Ex, an assessment is made of the GEOS modelled aerosol mass, vertical distribution, extinction, and particle size distribution. Section 2 discusses the in situ and remote sensing

instruments aboard the aircraft during CAMP²Ex that are crucial for detailing aerosol characteristics as well as the model simulations performed using GEOS. Results are presented in Section 3, from the perspective of the entire field campaign in Section 3.1 and for a case study using a flight transect through aged biomass burning aerosol in Section 3.2. Conclusions are given in Section 4 as well as recommendations for future development within GEOS and GOCART to improve future simulations of biomass burning aerosol.

**2 Data**

**2.1 Observations**

Based out of Luzon, Philippines, the NASA P3 aircraft completed 19 research flights throughout the period of 25 August 2019 through 5 October 2019. Flight tracks focused on two main regions, with the first half of the campaign concentrated near Luzon and southward into the Sulu Sea, while the second half included numerous flights to the north and

east over Philippine Sea. The P3 payload incorporated in situ and remote sensing instruments ideal for characterizing aerosols in addition to instrumentation for meteorology, clouds, precipitation, trace gases, and radiation (NASA ASDC, 2020). Here, we make use of the NASA Langley Aerosol Research Group Experiment (LARGE) suite of instruments, the Fast Integrated Mobility Spectrometer (FIMS; Kulkarni and Wang, 2006; Wang et al., 2017), and the 2nd generation High Spectral Resolution Lidar (HSRL2) (Burton et al., 2018). Chemical composition of the non-refractory submicron aerosol was provided by the

LARGE High Resolution Time-of-Flight Aerosol Mass Spectrometer (HR-ToF-AMS, Aerodyne Research, Inc.), while the optics array included nephelometers (TSI Inc., model 3563) and a particle soot absorption photometer (PSAP, Radiance Research) that provided scattering and absorption coefficients at three wavelengths, respectively. Ambient scattering was computed using the observed dry scattering, the growth factor fRH, and the ambient relative humidity. The mass concentration of black carbon was measured separately using the LARGE Single Particle Soot Photometer (SP2, Droplet Measurement

Technologies). Due to aerodynamic limitations of the sampling inlet, observations from the LARGE suite were only representative of particles smaller than 5 microns in aerodynamic diameter (McNaughton et al., 2006). There is a high uncertainty, up to 50%, in the aerosol mass concentrations observed by the AMS due to assumptions regarding the collection efficiency of different aerosol types. In addition, inconsistencies between measured mass concentrations and optical properties suggest the presence of significant submicron refractory mass (for which the HR-ToF-AMS is insensitive) or the potential for

particle losses within the tubing of the instrumentation leading to underestimates in aerosol mass. For these reasons, we have not used this dataset to quantitatively relate aerosol mass to radiative extinction through the mass extinction efficiency. Sea salt was assumed to be 3.27 times the mass concentration of sodium (Bian et al., 2019), measured by the Particle-Into-Liquid Sampler (PILS). To avoid contamination due to clouds, observations collected by LARGE were filtered using a cloud flag provided by SPEC (Stratton Park Engineering Company) cloud probes. Following both passive and active drying, the aerosol





size distribution was observed by FIMS at temporal resolution of 1 Hz for diameters ranging from 10 nm to 600 nm. The

HSRL2 provided aerosol backscatter and extinction at 355 nm, 532 nm, and 1064 nm. Mixed layer height was derived from

the HSRL2 backscatter (Scarino et al., 2014). One-minute averages were used for all observational data, apart from the HSRL2

aerosol backscatter observations which has a temporal resolution of 10 seconds.

CAMP$^2$Ex was accompanied by Propagation of Intraseasonal Oscillations (PISTON) field campaign which consisted

of a suite of observations collected aboard the Research Vessel Sally Ride (Chudler and Rutledge, 2021). The Sally Ride was

positioned in international waters to the northeast of the Philippines coincident with the time period of CAMP$^2$Ex flights. Here

we make use of the radiosonde launches from the cruise to classify the thermodynamic environment of the lower troposphere.

## 2.2 Model Simulations

During the CAMP$^2$Ex campaign, analyses and forecasts of meteorology and aerosols were provided by the GEOS

Forward Processing (FP) system, version 5.22. Since the campaign, numerous updates have been incorporated into GEOS

pertaining to the model physics, data assimilation, and the aerosol module. We have implemented these updates incrementally,

as summarized in Table 1, to determine the impact of each component on the simulation of aerosols during CAMP$^2$Ex. For

comparison to the observations, GEOS was properly sampled along the flight track using the one-minute average observational

files (Collow et al., 2020).

Within GEOS, aerosols are governed by the Goddard Chemistry Aerosol Radiation and Transport (GOCART) module

(Colarco et al., 2010). This module simulates the transport and optical properties of externally mixed hydroscopic and

hydrophilic organic and black carbon, sulphate, three size bins for nitrate, five size bins for sea salt, and five size bins for dust.

To implement updates and allow for future development, the (legacy) GOGART module code had been refactored and termed

"GOCART2G". GOCART2G includes a new radiatively active species, brown carbon, which is made possible through the

chemical transport of secondary organic aerosols. The optics look up tables for each aerosol species are the same as described

by Colarco et al. (2017). Sulphate, black carbon, brown carbon, and organic carbon are assumed to have a lognormal size

distribution with mode radii for dry particles of 0.0695 μm, 0.0188 μm, 0.0212 μm, and 0.0212 μm, respectively and a sigma

of 2.03, 2, 2.2, and 2.2 respectively. Aerosol optical depth (AOD) at 550 nm is constrained using the Goddard Aerosol

Assimilation System (GAAS) (Buchard et al., 2015; Randles et al., 2017). Apart from the "No GAAS" simulation, bias

corrected AOD observations from the Moderate Resolution Imaging Spectroradiometer (MODIS) aboard Terra and Aqua are

assimilated. GEOS 5.22 also assimilates AOD from the Aerosol Robotic Network (AERONET) (Holben et al., 1998).

Methodology for aerosol assimilation in GEOS is described in Section 3 of Randles et al. (2017). In brief, an analysis splitting

technique is used to obtain increments for the AOD, and the prognostic speciated aerosol mass is adjusted accordingly.

The underlying meteorology from GEOS is used for horizontal and vertical transport of the aerosol species, as well

as deposition and wind-driven emissions of dust and sea salt. Significant changes were made to the model physics beginning

with GEOS 5.25 that have direct impacts on aerosols (Arnold et al., 2020). The Chou-Suarez radiation scheme was replaced





with the Rapid Radiation Transfer Model for general circulation model applications (RRTMG; Norris et al., 2020). With regards to convection, a shallow convection scheme was introduced, and deep convection previously handled by the relaxed

Arakawa and Schubert (RAS) scheme was replaced by the Grell-Freitas (GF) parameterization (Freitas et al., 2020). Additionally, convective scale wet removal and transport of aerosol is now handled within the convective parameterizations instead of inside of GOCART (Arnold et al., 2020). The meteorology was constrained in two manners in which the data assimilation system (DAS) ran at the same time as the general circulation model (Online DAS), or the analysis produced from a previous simulation was used to reduce computational burden (Replay).


**Table 1. Model simulations performed with GEOS**

| Model Run | Data Assimilation | Model Physics | Anthropogenic Emissions | Biomass Burning Emissions | Aerosols | Resolution |
|---|---|---|---|---|---|---|
| GEOS 5.22 | Online DAS | RAS, No Shallow, Chou-Suarez (Rienecker et al., 2008) | HTAP v2v2 (Janssens-Maenhout et al., 2015), persisted 2012 | NRT QFED v2.5r1 (Darmenov and da Silva, 2015) | Legacy GOCART, AOD constrained | ~12 km with output saved at 0.25°, 72 vertical levels |
| GEOS 5.25 | Online DAS | GF, UW Shallow, RRTM-G (Arnold et al., 2020) | HTAP v2v2 (Janssens-Maenhout et al., 2015), persisted 2012 | NRT QFED v2.5r1 (Darmenov and da Silva, 2015) | Legacy GOCART, AOD constrained | 0.25°, 72 vertical levels |
| GOCART2G | Replay to GEOS 5.25 | GF, UW Shallow, RRTM-G (Arnold et al., 2020) | CEDS 2019 (https://doi.org/10.25584/PNNLDataHub/1779095) | QFED v2.5r1 (Darmenov and da Silva, 2015) | GOCART2G, AOD constrained | 0.25°, 72 vertical levels |





| No GAAS | Replay to MERRA-2 (Gelaro et al., 2017) | GF, UW Shallow, RRTM-G (Arnold et al., 2020) | CEDS 2019 (https://doi.org/10.25584/PNNLDataHub/1779095) | QFED v2.5r1 (Darmenov and da Silva, 2015) | GOCART2G, free running aerosols | 0.5°, 72 vertical levels |
|---|---|---|---|---|---|---|

## 3 Results

### 3.1 Campaign-Wide

#### 3.1.1 Lower Troposphere Meteorology

As discussed in Section 2.2, upgrades were made to the model physics, particularly the convection, that impacts the vertical transport of aerosols, wet deposition, and their extinction through changes in the relative humidity. Lower tropospheric temperature and humidity before and after the changes were implemented are compared to the PISTON sondes in Figure 1. A cool bias is present in both GEOS 5.22 and GEOS 5.25 in the lowest 4 km. While some improvement in the bias can be seen between 2 and 4 km in GEOS 5.25, it is evident that there is a degradation in temperature below 2 km with the updated model physics (Figure 1a). The same is true for the vertical profile of specific humidity. A dry bias was greatly improved above 1 km in GEOS 5.25; however, the dry bias became exacerbated near the surface (Figure 1b).

The diurnal evolution of the planetary boundary layer (PBL) and lower troposphere is evaluated in panels c through f of Figure 1 using relative humidity. Relative humidity was selected for this evaluation since it is used in the optics lookup tables for aerosols. The Philippines are eight hours ahead of coordinated universal time such that 0z (Figure 1c) represents a morning profile while 12z (Figure 1e) represents an evening profile. GEOS 5.25 has difficulty capturing the inversion that develops during the daytime hours of 12z and 18z. While there is a hint of an inversion in GEOS 5.22, it is located too high. This is likely due to deficiencies in the turbulence parameterizations in GEOS as well as the coarse vertical resolution.

#### 3.1.2 Aerosols

The temporal evolution of aerosols in the Philippines region is evaluated using observations of daily mean AOD from two AERONET stations in the area in Figure 2. Located on the island of Luzon, Manila served as the base of operations for the campaign, while Tai Ping is an island within the South China Sea. Both stations reported AOD at 500 nm. The Angstrom exponent for 440 nm and 675 nm was used to convert to the AOD at 550 nm for comparison to GEOS. Routine observation of AOD in the region is a challenge due to frequent cloudiness, and this resulted in a lack of observations in Manila prior to late





September. It is possible that the larger values of AOD plotted for Manila are biased high due to cloud contamination as this comparison uses the Level 1.5 AERONET product, or localized urban emissions not in the CEDS emissions dataset.


Profiles of aerosol backscatter below the aircraft were collected along the flight paths by the HSRL2 at three wavelengths: 355 nm, 532 nm, and 1064 nm. While three of the model simulations constrain the AOD at 550 nm using bias corrected MODIS observations (Randles et al., 2017), the vertical profile of aerosol mass and backscatter are prognostic fields. For reference, mixed layer height (MLH) from the HSRL2 and PBL height from GEOS have been added to Figure 3. The

native terminology for the data products, MLH and PBLH have been retained to reinforce that these quantities are not computed in the same manner. The height of the PBL is too high in GEOS as confirmed by the profiles of relative humidity in Figure 1 and the height of maximum backscatter in Figure 3. The three versions of GEOS with AOD assimilation are indistinguishable and for this reason, only the final GOCART2G simulation is show in Figure 3. GEOS suffers from a negative bias in aerosol backscatter above the boundary layer, between 1 and 2 km, and a positive bias at the top of the boundary layer. This is trend

is consistent at all three wavelengths, but most apparent at 355nm (Figure 3a). Within the boundary layer itself, the agreement between GEOS and the observations is wavelength dependant. The model does not have enough backscatter at 355 nm yet there is too much backscatter at 1064 nm. This indicates there is either an underlying error in the aerosol speciation or within the GOCART optics tables, but the wavelength-dependent bias likely points to the particle size distribution as the underlying discrepancy. Subtle differences in the backscatter are present between the GEOS versions. While the differences may not be

statistically significant, the combination of physics and aerosol updates made in GEOS 5.25 and GOCART2G resulted in a slight improvement in backscatter within the PBL at 355 nm and 1064 nm.

Aerosol extinction was derived from the molecular channel signal as described by Hair et al. (2008). Although the results in Figure 4 are qualitatively similar to backscatter, additional information can be gained by analysing extinction. With respect to extinction at the two shorter wavelengths, there is a larger impact of the change in relative humidity between GEOS

5.22 and GEOS 5.25 than the aerosol updates implemented in GOCART2G. At 355 nm and 532 nm, the maximum extinction within the column at the 75th percentile for GOCART2G compares for well to the 75th percentile for the HSRL2, although the peak extinction is located too high due to the height of the boundary layer in the model. Although the median is overestimated, there is some benefit of the updates made in GOCART with respect to aerosol extinction. It is also evident that the lidar ratio differs with and without the assimilation of AOD, indicating there is some impact of the AOD assimilation on the aerosol

speciation.

A more in-depth assessment of aerosol extinction can be made by filtering the 532 nm extinction by the HSRL2 derived aerosol type (Burton et al, 2012). Five aerosol types are considered based on the region of interest: marine, polluted marine, smoke, fresh smoke, and urban pollution. The sample size for each aerosol type can be found in the supplemental document, as well as the GEOS aerosol speciation for each HSRL2 derived aerosol type. Due to a drastic decrease in the

sample size above 2 km, only the bottom 2 km are shown in Figure 5. There is also a focus placed on the GOCART2G and No GAAS simulations due to the previously noted similarities among the GEOS runs. Unsurprisingly, smoke stands out as



having the largest extinction (Figure 5e), however this could also be related to the fact that it has the smallest sample size of the aerosol types. Smoke is also responsible for a negative bias in GEOS, and the largest difference between the runs with and without the assimilation of AOD through GAAS. This could indicate deficiencies in the model's optical properties for smoke,

the transport, meaning the smoke plume is not in the correct location without the data assimilation, or uncertainties in the emissions. The vertical profile in extinction for fresh smoke and urban pollution are similar perhaps since the HSRL2 can have difficulty distinguishing between the two (Figure 5c and d). This could be the case between smoke and urban pollution as well, as indicated by the consistent model biases between the two aerosol types with a slight underestimation of extinction within the boundary layer. Given that there is a slight positive bias in GEOS when all aerosol types are considered, it is worth further

investigating the cancellation of errors from marine, biomass burning, and urban aerosols.

Unlike the remote sensing capabilities of the HSRL2, the LARGE optical array is in situ and can provide a direct comparison between extinction and aerosol composition. This comes at the cost of a much smaller data sample that is only representative of fine particles that efficiently sampled by the inlet. The modelled aerosols were subsampled such that only particles small enough to be observed, with an aerodynamic diameter less than 5 µm, were included in the extinction and

scattering calculations for comparison to LARGE. As a result of the smaller sample size, the vertical profile for median 532 nm extinction is not as smooth, extinction within the boundary layer is noticeably smaller (Figure 6a). An evaluation of individual flights demonstrated that agreement between the in situ and remotely sensed extinction was better on flights that captured biomass burning aerosol, likely because the composition was dominated by fine particles as opposed to coarser nitrate, sea salt, and dust (not shown). Results for the LARGE in situ extinction are consistent with the HSRL2 comparison. All GEOS

versions underestimate extinction around 2 km and overestimate extinction at the top of the boundary layer, however this overestimation extends down to the surface.

The contribution of relative humidity to the overestimation of extinction in the boundary layer can be assessed through the dry extinction in which the aerosols are dried to at least 40% relative humidity before being passed to the optical array. Dry extinction in GEOS 5.22 is in excellent agreement with the observations, though GEOS 5.25 and GOCART2G also

perform well (Figure 6a). On the contrary, GEOS overestimates 532 nm extinction under ambient conditions (Figure 6b). Another way to investigate the role of relative humidity in GEOS is to bias correct the relative humidity by running the GOCART optics code using the model's aerosol mass concentration but replacing the relative humidity with what was observed by the aircraft (Figure 6c). Except for a decrease in the extinction at the top of the PBL in GEOS 5.22 and a small increase in extinction in all GEOS runs, there is little change in the representation of extinction through correcting the relative

humidity. This is not limited to GEOS as minimal improvement occurred through correcting RH biases in the Navy Aerosol Analysis and Prediction System Reanalysis (NAAPS-RA) (Edwards et al., 2021). These results suggest that the discrepancy in ambient extinction is a result of model treatment of particle hygroscopicity, and less dependent on aerosol concentration (i.e., loading) or relative humidity.

Not only is the total aerosol mass concentration overestimated in GEOS, but the speciation also disagrees with the

LARGE observations (Figure 7). GEOS greatly overestimates black carbon in the lowest 4 km (Figure 7a). While the mass





concentration of black carbon was reduced above the boundary layer by instituting the convection updates in GEOS 5.25, it led to an additional build up on black carbon in the boundary layer. A beneficial reduction in black carbon occurred with the GOCART2G updates, a direct result of a lower scaling factor for the biomass burning emissions. The impact of the assimilation of AOD can be seen by comparing the lines for GOCART2G and No GAAS. Above the boundary layer, the two runs are

essentially the same. Without GAAS turned on, the black carbon is already excessive, yet the AOD for the column is too low. This results in positive values for the analysis increment for black carbon mass and a doubling of the positive bias in the mass concentration within the boundary layer. A similar deficiency, with the same explanation, is shown for organic aerosol (Figure 7b). Though denoted as organic carbon, GEOS represents this as organic matter by using a multiplicative factor of 1.8. Unlike black carbon, GEOS performs well in terms of the amount of organic carbon above the PBL. A notable increase in organic

carbon is present in the boundary layer in GOCART2G. Since brown carbon originates as a portion of what was organic carbon prior to GOCART2G, it is being included as organic carbon in the figure.

Sulphate and nitrate suffer from the opposite problem (Figure 7c and d). In general, there is not enough aerosol for these two species in the model. A comparison of the observed profiles for organic carbon, sulphate, and nitrate suggests multiple sources and air masses containing the aerosols throughout the CAMP$^2$Ex campaign. The boundary layer tends to be

influenced by biomass burning aerosol, particularly during the first half of the campaign prior to the monsoon transition, while the lower free troposphere contains anthropogenic aerosol transported from East Asia (Hilario et al., 2021). It is evident that the improvement in sulphate near the surface in GEOS 5.25 was matched by a degradation just above the top of the boundary layer due to a change in the vertical transport. Unfortunately, the same reduction in biomass burning emissions that assisted with the mass of carbon in GEOS with GOCART2G, also led to a reduction in sulphate. There is an underestimation in fine

mode nitrate within the entire profile shown in Figure 6d. While deficiencies in other processes are possible, one explanation could be that the model is skewing more towards coarse mode nitrate, consistent with the biases in 1064 nm extinction in Figure 3. To match the inlet size, only the smallest size bin is included here. There is also the potential that sulphate and nitrate produced over mainland Asia are excessively scavenged prior to reaching the Philippines region or deficiencies in precursor species like ammonium.


## 3.2 Case Study along a Smoky Transect

For a more detailed look at a biomass burning plume, a segment with roughly constant altitude from research flight (RF) 9 was selected as indicated by the flight map in Figure 8. The PSAP data is prone to uncertainty when the aircraft performs vertical profiling manoeuvres, making the consistent altitude ideal for absorption data. During the central part of the segment,

the aircraft was well within the boundary layer (Figure 9). Data points just before and after this portion were also included to investigate deviations in aerosol extinction due to relative humidity. For this section, only the final model run, GOCART2G, including all model updates and the assimilation of AOD is considered. Since the main goal of this section is to evaluate the aerosol intensive properties, it is irrelevant which model simulation is used as the optics look up tables are unchanged. While





the aerosol mass concentration and relative humidity have the potential to differ in each of the model simulations, the
relationship between the two and the optical properties remain the same.

The observed aerosol composition during this flight segment was predominantly organic carbon (81.3%) though it
should be noted that the SP2 malfunctioned during this flight and the concentration of black carbon is not available. GEOS
represented that percentage of organic carbon well however struggled with the relative concentrations of sulphate, nitrate, and
sea salt. The ratio of sea salt is exceptionally high in GEOS, despite the multiplication factor to convert observed sodium to
sea salt. There are a few possible explanations for this overestimation. The total mass of sea salt in GEOS could be correct and
the bias detected here could be related to the assumed size distribution. Only the three finest sea salt bins were included to
match the inlet size for the aircraft. However, given the preference for coarse mode sea salt in GEOS (Bian et al., 2019), we
suspect this is not the case. A more likely scenario is that the AOD assimilation increases sea salt instead of sulphate and nitrate
as those two species are not as prevalent. The deficiency in sulphate and nitrate was noted throughout the entire CAMP²Ex
campaign (Figure 7c and d).

Table 2. The percent contribution of aerosol species to the total aerosol mass during the RF9 flight segment from observation and
GEOS GOCART. Black carbon is not available from the observations for this flight segment but the ratio of BC:OC in GEOS is
0.0677:1.

|  | LARGE Observations | GEOS |
|---|---|---|
| Organic Carbon | 81.5% | 80.0% |
| sulphate | 13.1% | 6.8% |
| Sea Salt | 2.8% | 13.2% |
| Nitrate | 2.4% | 0.0% |


Dry and ambient aerosol optics for the flight segment are display in Figure 10. The top two panels are coloured based
on the bias within GEOS for the mass concentration of organic carbon, which is always positive, and are representation of dry
conditions. The smallest biases in organic carbon occur when both the observations and GEOS indicate lower amounts of dry
scattering and extinction (Figure 10a and b). It is evident that GEOS needs a large bias in the mass concentration of organic
carbon to accurately represent dry extinction. Overall, there is a mean negative bias in dry scattering and extinction despite
the positive bias in aerosol mass. Under ambient conditions, GEOS performs well with respect to extinction for many of the
data points, except for a cluster of data points where LARGE observes an extinction of ~0.5 km⁻¹ yet GEOS has up to triple
the extinction (Figure 10c). As indicated by the blue shading for those data points, GEOS is too humid when the overestimation
in extinction at 532 nm occurs. Some improvement can be seen when extinction is computed for GEOS using the observed
relative humidity, however there is still an overall positive bias in ambient extinction (Figure 10d), indicating a concern with
hygroscopicity in the model.





A comparison of the range in values for the dry and ambient extinction reveals that there is minimal hygroscopic growth in the observations while GEOS extinction is highly sensitive (Figure 10b and 10d). There is a hygroscopic growth factor, fRH (computed using 40% and 80% relative humidity), of less than one in the observations, such that the aerosol shrinks rather than swells with increasing humidity. The average fRH for the flight segment is 0.915. In GEOS, the average fRH is 2.16 and anything less than one would be considered unphysical and is not permitted in the current optical look-up tables. Laboratory studies and aircraft observations of fresh smoke demonstrate a range of fRH values depending on fuel type, fire conditions, and RH (Day et al., 2006), with an fRH typically between ~1 and ~2 (Hand et al., 2010). The review of previous studies presented by Hand et al. (2010) shared conflicting results regarding aged smoke with one study on par with fresh smoke and the another suggesting higher values of fRH for aged smoke. Like during the smoke transect from CAMP²Ex, other field campaigns have observed an fRH below 1 on select occasions, hypothesized to be due to particle restructuring (Shingler et al., 2016). Even though a value of fRH below 1 is not possible in GEOS, 2.16 is still too high based on prior estimates in the literature.

Previous studies have documented a wavelength dependence on the SSA for biomass burning aerosol, and uncertainty in observations of SSA in smoke plumes (Pistone et al., 2019). We find that the SSA is underestimated in GEOS and the modelled spectral dependence is not in good agreement with the observations. Ambient SSA is available at 550 nm from the LARGE observations and is displayed in Figure 11 in comparison to GEOS for the flight segment. Nearly all points for the observations have a SSA greater than 0.98 while GEOS indicates a SSA below 0.96. There is also more variability with the SSA in GEOS connected to the dependence on RH. Correcting the bias in RH does not improve the overall mean value of ambient SSA at 550 nm, however it does decrease the variability. Mean values of dry SSA across the entire flight transect are shown in Figure 12 for three wavelengths: 450 nm, 550 nm, and 700 nm. As previously indicated for ambient RH, the mean SSA in GEOS is too absorbing at 550 nm, and this is also the case for dry SSA at the three wavelengths (Figure 12). The observations indicate a linear relationship between SSA and wavelength, which is not the case for GEOS as the model is excessively too absorbing at 450 nm.

### 3.3 Aerosol Size Distribution

Through changing the optics lookup tables, a greater extinction could be achieved in GEOS by altering the assumed particle size distribution for organic and brown carbon such that the mean radius is larger and the width of the distribution is wider. We begin by looking at two flights, RF9, the flight examined in Section 2.2, and RF10, which also captured aged biomass burning aerosol but downstream in the Philippine Sea on the following day. Both flights are filtered to only include timesteps in which the chemical influence flag indicated biomass burning aerosol (doi: 10.5067/Airborne/CAMP2Ex_TraceGas_AircraftInSitu_P3_Data_1). RF9 contained 80% organic carbon and 12.5% sulphate, while RF10 had 70% organic carbon, 14% sulphate, and a non-negligible content of sea salt (15%). Figure 13 shows the observed aerosol size distribution from FIMS as well as the assumed sized distribution for organic carbon and sulphate in



GEOS. The size distributions for GEOS have been scaled such that the peak of the lognormal distribution matches the maximum from the observations. Comparing the range of the y-axis for RF9 and RF10, it is obvious that RF9 had a higher aerosol loading as the aircraft sampled the Sulu Sea region, closer to the source of the smoke. The observations indicate that biomass burning aerosol in this region has a bimodal size distribution, a feature that has been known for over a decade (Reid et al., 2005). Neither of the observed peaks line up with the assumed size distribution in GEOS for organic carbon or sulphate,

with both peaks in GEOS falling in between the observed peaks.

Agreement in peak radius is somewhat better for RF10; however, the width of the observed distribution, particularly for the peak with a smaller geometric radius, is narrower than what is assumed by GEOS. The primary peak in the size distribution is shifted towards a larger radius. The relative magnitude of the two peaks in the bimodal distribution in the observed size distributions is intriguing as there is comparatively less aerosol peaking around 0.1 µm. If we were to assume

that the smaller peak radius in the FIMS observations corresponds to organic carbon, the mean radius in GEOS should be reduced. This, however, contradicts earlier findings that the extinction in GEOS should be increased, unless of course, a corresponding increase in the mean radius for the relatively minor content of sulphate offsets a smaller radius for organic carbon. Furthermore, Chin et al. (2009) reported the effective radius for organic matter in GOCART is likely too small.

Figure 14 provides a closer look at the aerosol size distribution for the flight segment from RF9 evaluated in Section

3.2, which yields slightly different results, indicating variability in the observed particle size distribution for biomass burning aerosol during an individual flight. There is not, however, variability in the size distribution within the smoke transect. The primary observed peak from Figure 13 is not present. There is likely internal mixing occurring during the lower-altitude flight segment, which cannot be achieved within GEOS. A very similar peak radius for biomass burning aerosol has been observed during SEAC[4]RS and ORACLES (Schill et al., 2021), as denoted by the dashed line in Figure 14, however observations from

the other campaigns indicate a narrower distribution. Based on these results, it can be hypothesized that by using the Schill distribution for the brown carbon component of the aerosol to mimic internally mixed, aged aerosol, an improvement will occur in the overall representation of biomass burning aerosol in GEOS.

## 4. Conclusions

The CAMP[2]Ex field campaign, comprised of 18 research aircraft flights using NASA's P3, took place from late August 2019

to early October 2019. The aircraft collected observations of clouds, aerosols, radiation, and meteorology using a suite of in situ and remote sensing instruments. Here, we take advantage of the plethora of data to evaluate the representation of biomass burning aerosols in GEOS. This was a timely exercise as recent updates in the convective parameterizations and well as GOCART had the potential to alter the vertical profile of aerosol mass and extinction. Updates of particular importance for evaluation included the introduction of brown carbon, a switch from a relaxed Arakawa and Schubert convective

parameterization to the Grell-Freitas convective parameterization coupled with the University of Washington shallow convective scheme and allowing convective scavenging to occur within the moist parameterization instead of within





GOCART. Model updates were evaluated incrementally to determine their individual impacts, however many biases noted in GEOS are independent of these changes.

The findings of this study highlight areas of focus for future development within GEOS and GOCART. From a model
physics perspective, there is a need for improvement in the turbulence and shallow convection schemes that govern the height of the PBL and the vertical transport as indicated by the vertical profiles of aerosol mass concentration as well as atmospheric moisture. Using aerosol mass concentration as a tracer indicates an overestimate of aerosol within the boundary layer, with mass piling up at the top of the PBL, unable to sufficiently penetrate into the free troposphere. A similar finding was reported by Bian et al. (2021) when investigating organic aerosol in GEOS using aircraft observations in Brazil. While there was some
improvement in relative humidity above 2 km through new convection parameterization in GEOS, biases remain below this height, in addition to the inability to capture the inversion. With a limited sample size associated with the aircraft flights, it is difficult to evaluate the vertical profile of aerosol mass across the diurnal cycle. It is, however, recommended that future studies investigate the diurnal cycle of aerosol backscatter using a stationary lidar, such as the University of Wisconsin HSRL aboard the Sally Ride during the PISTON campaign, based on differing biases in relative humidity across the diurnal cycle.

Deficiencies in sulphate and nitrate emphasize the need for assessing the budget for urban aerosols. Most of the sulphate and nitrate in GOCART is not emitted as aerosol, but rather forms through aqueous oxidation and heterogenous chemistry of their precursory gaseous species. The production of nitrate and sulphate within GOCART should be reassessed, and there is also the possibility that too much of these species is being removed from the atmosphere through sedimentation, dry deposition, wet deposition, and/or convective scavenging. This is, however, dependent on an accurate emissions inventory
for Asia within the CEDS database. It is plausible that the CEDS inventory lacks the proper emissions for precursor species for sulphate and nitrate over China, leading to an underestimate of aerosol mass downstream.

A more in-depth focus was placed on biomass burning aerosol. A limitation of GOCART is that the aerosol is considered externally mixed. Smoke is known to be internally mixed (Reid et al., 2006) and there is evidence of this in the FIMS observations from CAMP$^2$Ex. An abundant quantity of black and organic carbon is present in GEOS, which was also
found to be the case during the ORACLES campaign (Shinozuka et al., 2020). The assimilation process constrains extinction for the whole column but does not place any constraint on aerosol mass, exacerbating an already existing bias in the mass of carbon. At the root of the overestimates of carbonaceous aerosol is an underestimate in the extinction defined through optics look up tables as a function of mass, humidity, and the mode radius and width of a lognormal distribution representing the aerosol size distribution. A comparison to in situ observations from the CAMP$^2$Ex campaign demonstrated issues with both
the dependency on humidity and particle size distribution. While it is not possible for GEOS to match the hygroscopic growth from the observations, it is evident that GEOS has too large of an increase in extinction with an increase in humidity. Some of the concern with the particle size distribution could be rectified by adjusting the assumed particle size distribution for brown carbon to have a larger mode radius or a wider distribution. A comparison of the assumed particle size distribution to observations from FIMS indicated that the current mode radius is too small. The observed mode radius is in excellent
agreement with observations from past field campaigns, however the width for CAMP$^2$Ex is larger. Modifying the assumed





particle size distribution for brown carbon would need to be thoroughly evaluated as it would also impact fields such as the single scattering albedo and Angstrom exponent.

**Data Availability**

GEOS model output sampled along the aircraft trajectories for CAMP²Ex are available for download through the NCCS Dataportal at https://portal.nccs.nasa.gov/datashare/iesa/campaigns/CAMP2EX/. CAMP²Ex observational datasets are available at https://asdc.larc.nasa.gov/project/CAMP2Ex.

**Acknowledgements**

Computational resources supporting this work were provided by the NASA High-End Computing (HEC) Program through the NASA Center for Climate Simulation (NCCS) at Goddard Space Flight Center. This work was supported by the NASA Earth Science Project Office (ESPO)'s CAMP²Ex Mission. We would also like to thank Jeff Reid for his support and useful discussions, Josh DiGangi for providing the chemical influence flag, and Jian Wang, the PI for FIMS.

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







**Figure 1: Vertical profile of (a) temperature and (b) relative humidity biases with respect to all PISTON sondes relative humidity at (c) 0z, (d) 6z, (e) 12z, and (f) 18z from the PISTON sondes, GEOS 5.22 and GEOS 5.25.**





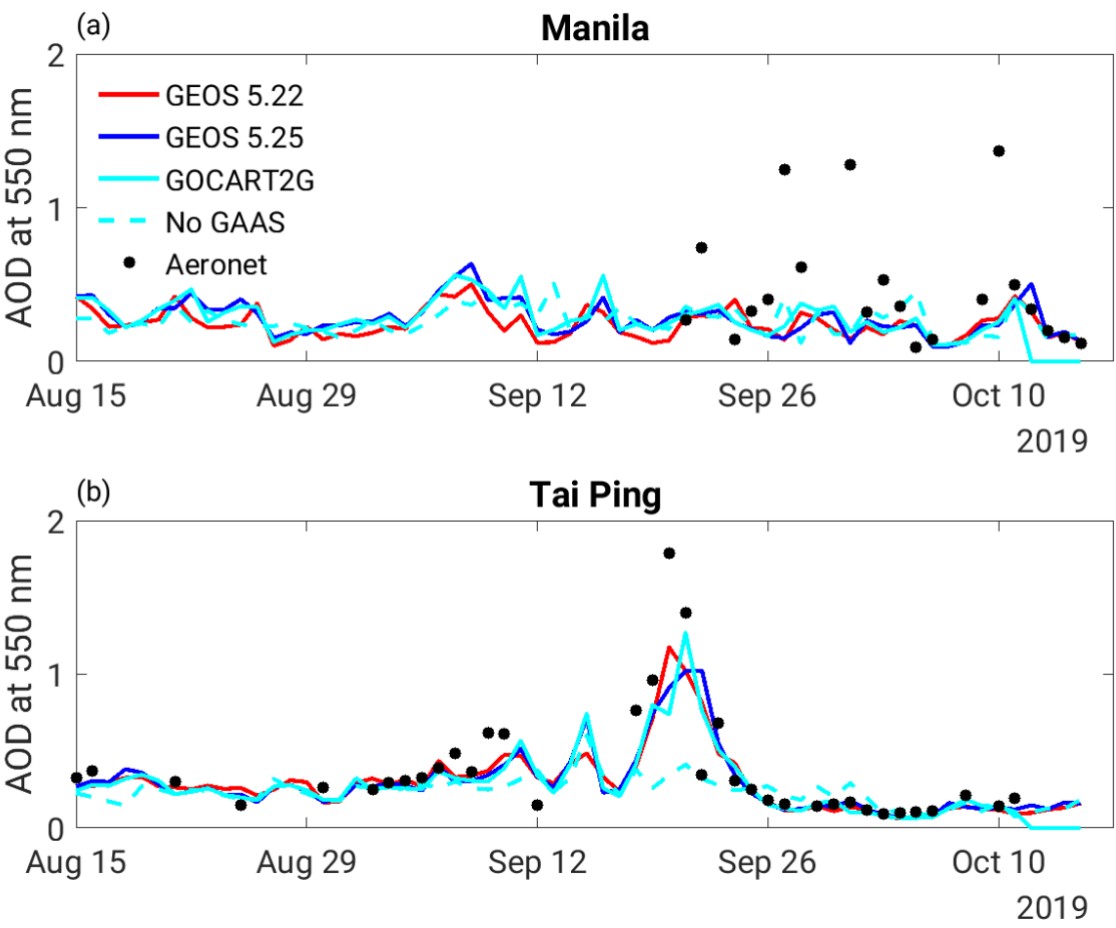

Figure 2. Timeseries of aerosol optical depth at 550 nm from AERONET sites in (a) Manila, Philippines and (b) Tai Ping Island and the corresponding timeseries from GEOS 5.22, GEOS 5.25, GOCART2G, and GOCART2G without GAAS.





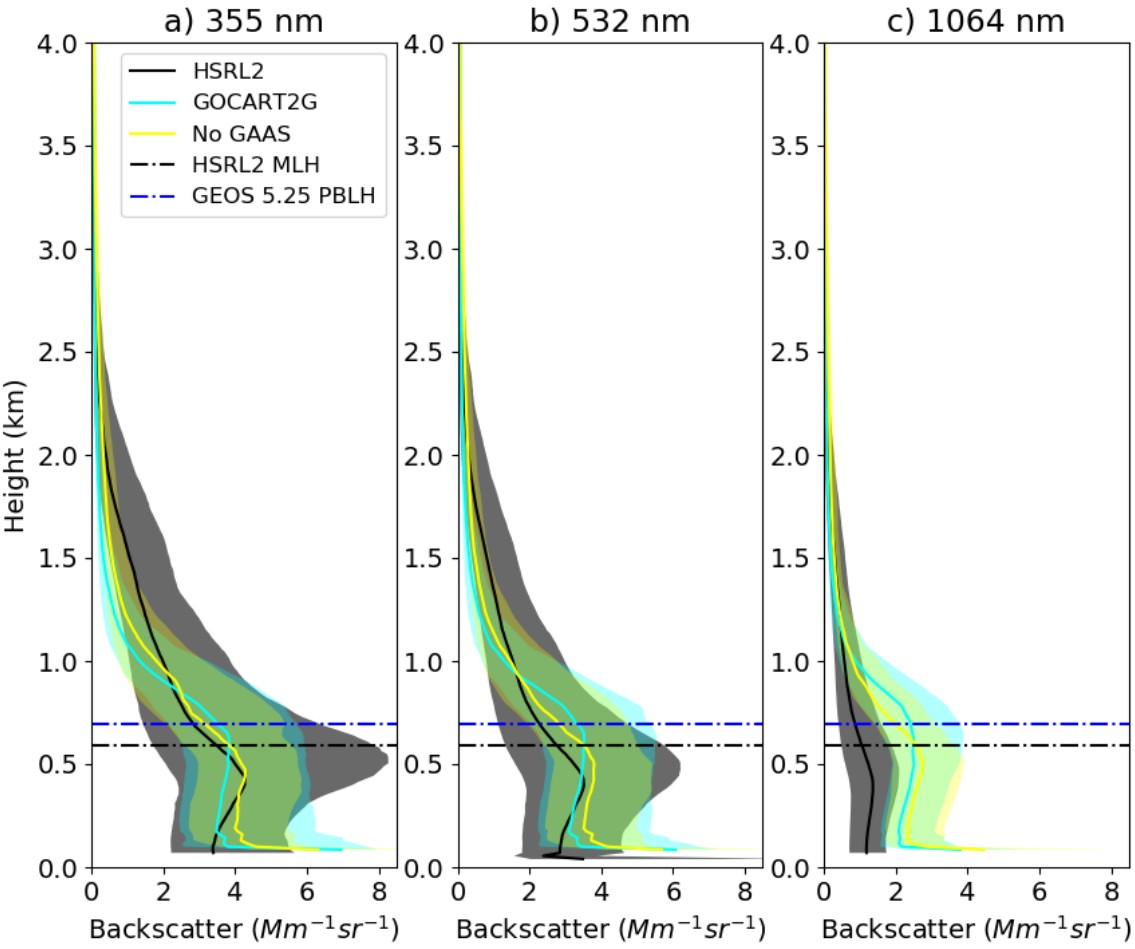

**Figure 3: Median backscatter at (a) 355 nm, (b) 532nm, and 1064 nm during all research flights from the HSRL2, GOCART2G, and GOCART2G without aerosol assimilation (No GAAS). Profiles are shaded between the 25th and 75th percentiles. Mixed layer height (MLH) from the HSRL2 and planetary boundary layer (PBL) height in GEOS are added for reference as dashed lines. Note that PBL height for GOCART2G is the same as GEOS 5.25. An analogous figure with all model simulations can be found in the supplemental document.**



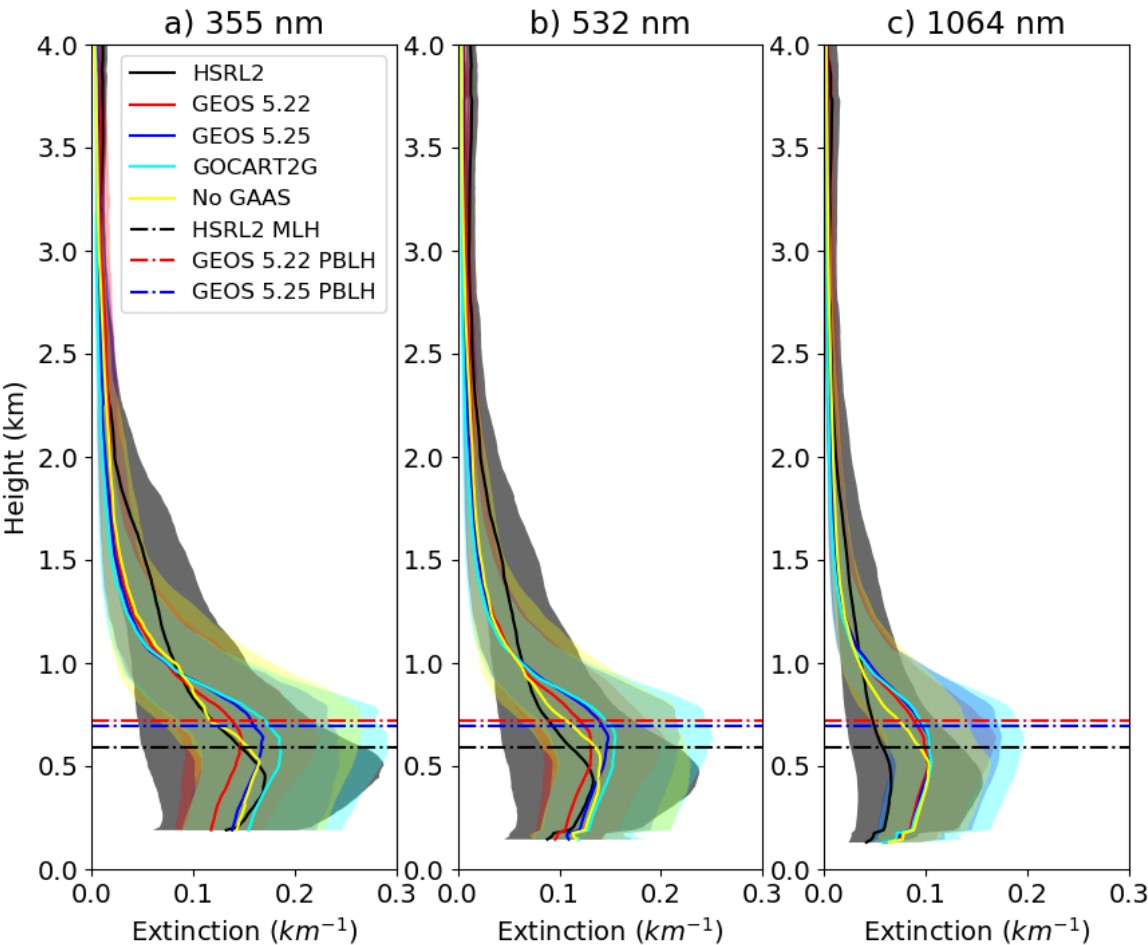

**Figure 4. Median extinction at (a) 355 nm, (b) 532nm, and 1064 nm during all research flights from the HSRL2, GEOS 5.22, GEOS 5.25, GOCART2G, and GOCART2G without aerosol assimilation (No GAAS). Profiles are shaded between the 25th and 75th percentiles. Mixed layer height (MLH) from the HSRL2 and planetary boundary layer (PBL) height in GEOS are added for reference as dashed lines. Note that PBL height for GOCART2G is the same as GEOS 5.25.**





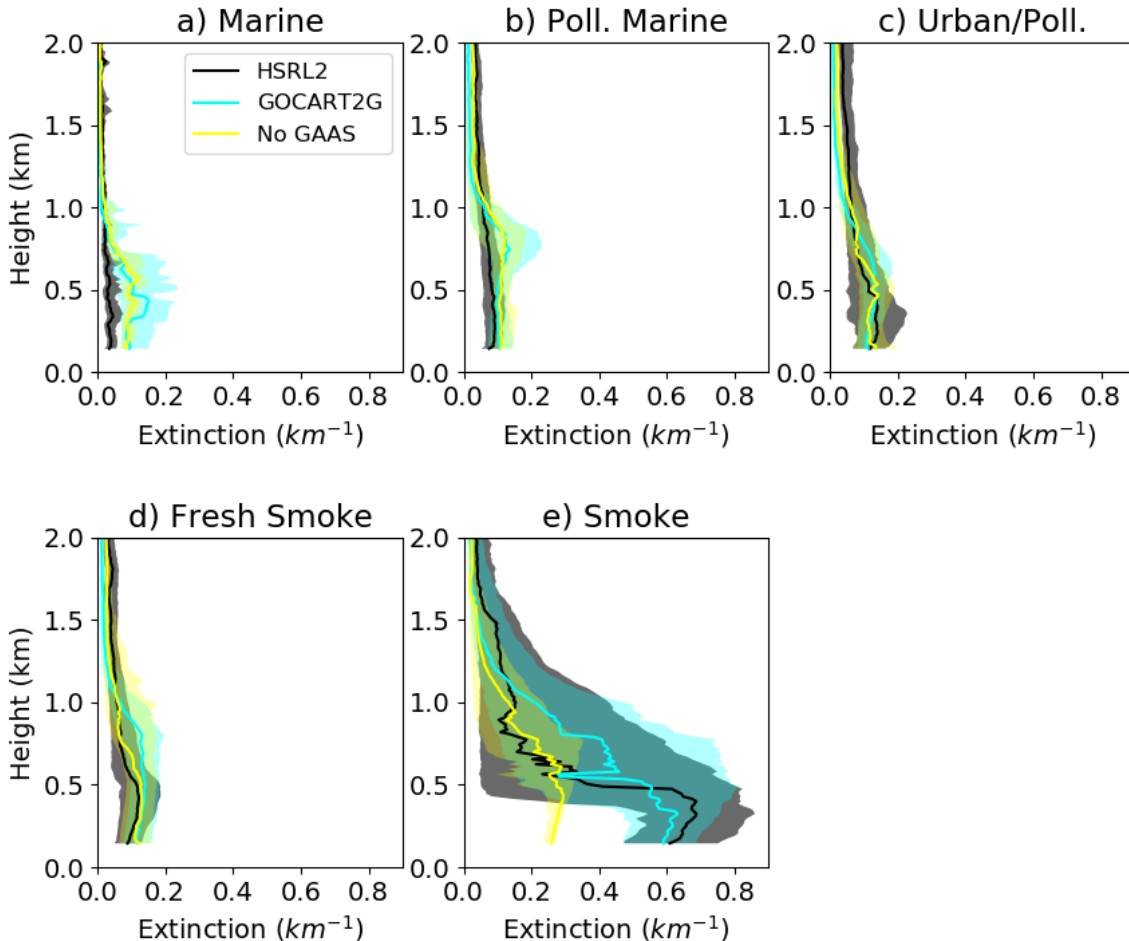

**Figure 5. Median extinction at 532 nm during all research flights from the HSRL2, GOCART2G, and GOCART2G without aerosol assimilation (No GAAS) filtered based on the HSRL2 aerosol id for (a) marine, (b) polluted marine, (c) urban pollution, (d) fresh smoke, and (e) smoke aerosols. Profiles are shaded between the 25th and 75th percentiles. An analogous figure with all GEOS simulations can be found in the supplemental document.**





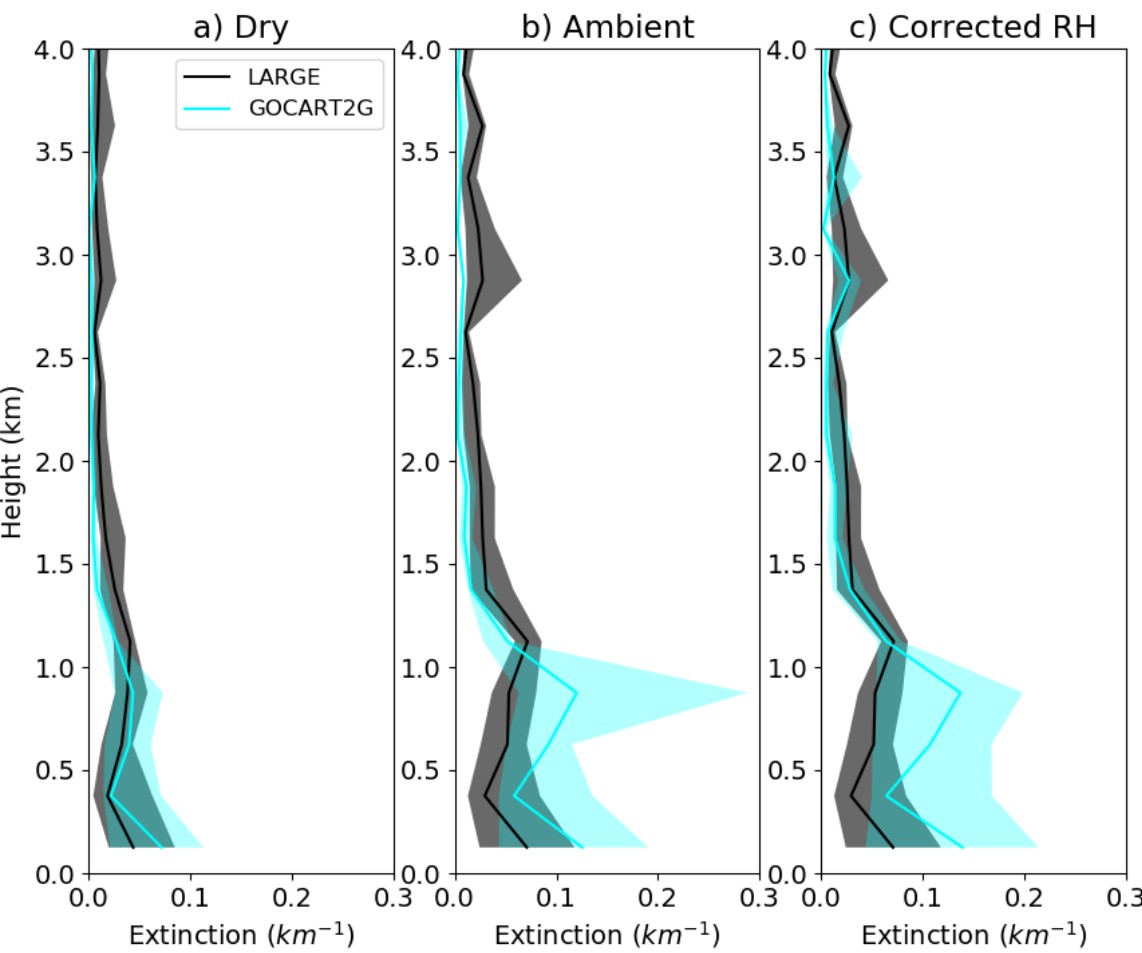

**Figure 6. Median extinction at 532 nm for all research flights from LARGE and GOCART2G for (a) ambient, (b) dry, and (c) observation corrected relative humidity. Profiles are shaded between the 25th and 75th percentiles. An analogous figure with all model simulations can be found in the supplemental document.**





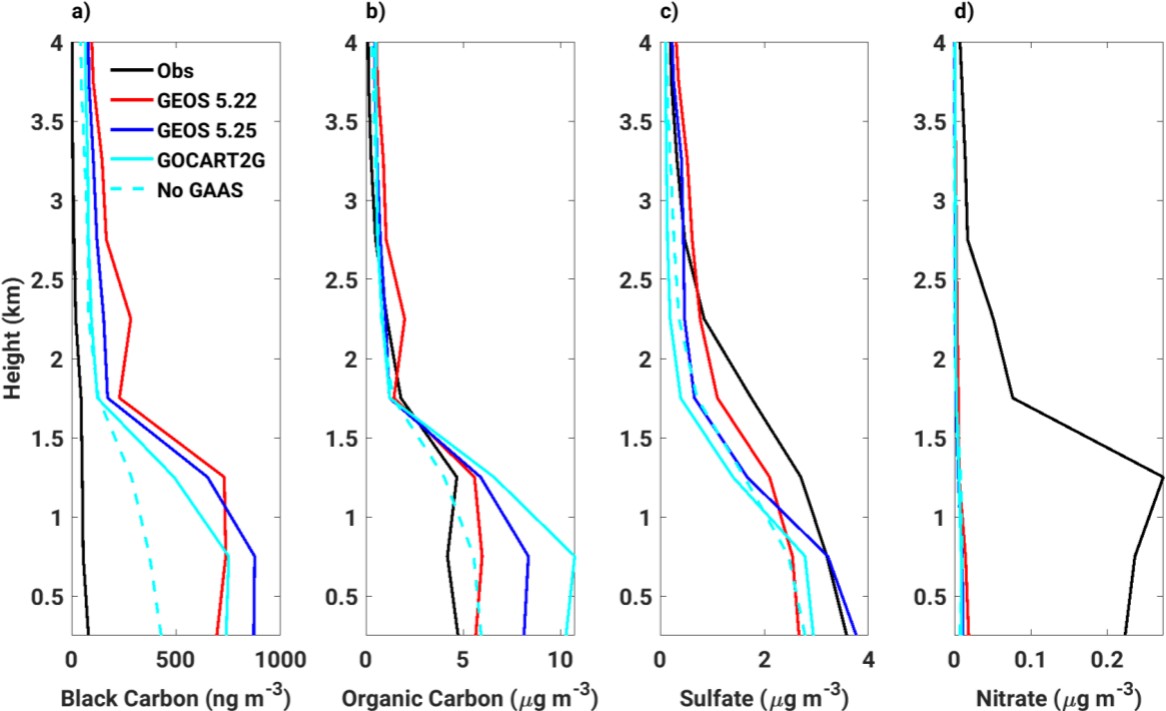

**Figure 7. Mean vertical profile of aerosol mass concentration from the LARGE observations, GEOS 5.22, GEOS 5.25, GOCART2G, and GOCART2G without GAAS for (a) black carbon, (b) organic carbon, (c) sulphate, and (d) nitrate**

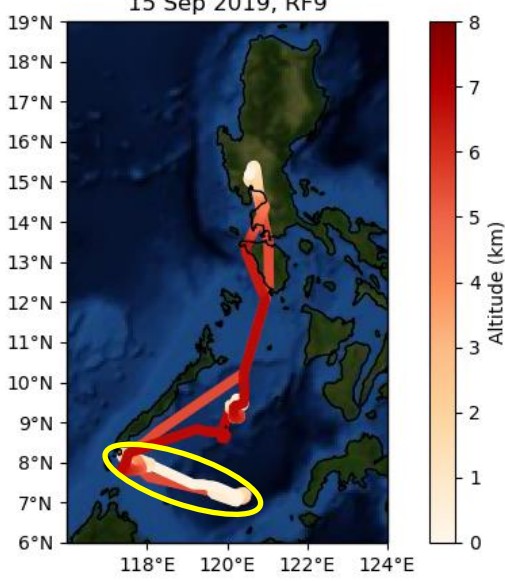

**Figure 8: Flight trajectory for RF9 on 15 September 2019 coloured by the altitude of the aircraft. The yellow oval indicates the low level transect focused on in Section 3.2.**



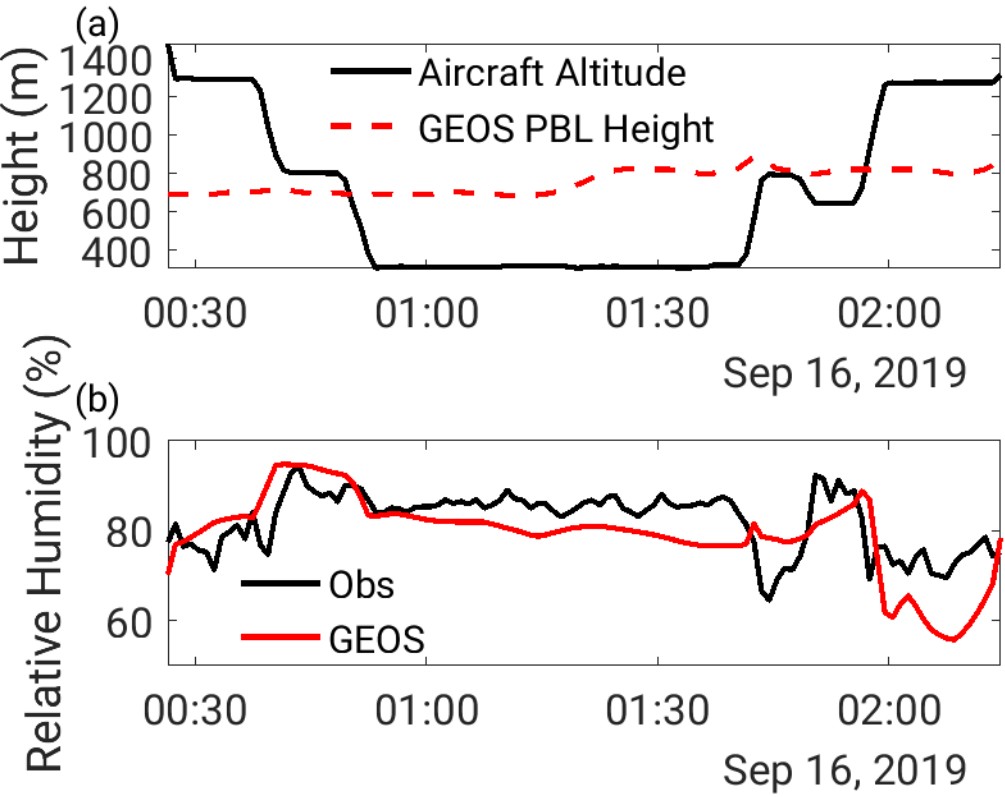

**Figure 9: (a) Height of the aircraft during the smoke transect from RF9 in comparison to the height of the planetary boundary layer in GEOS and (b) the relative humidity from the observations and GEOS during the transect.**



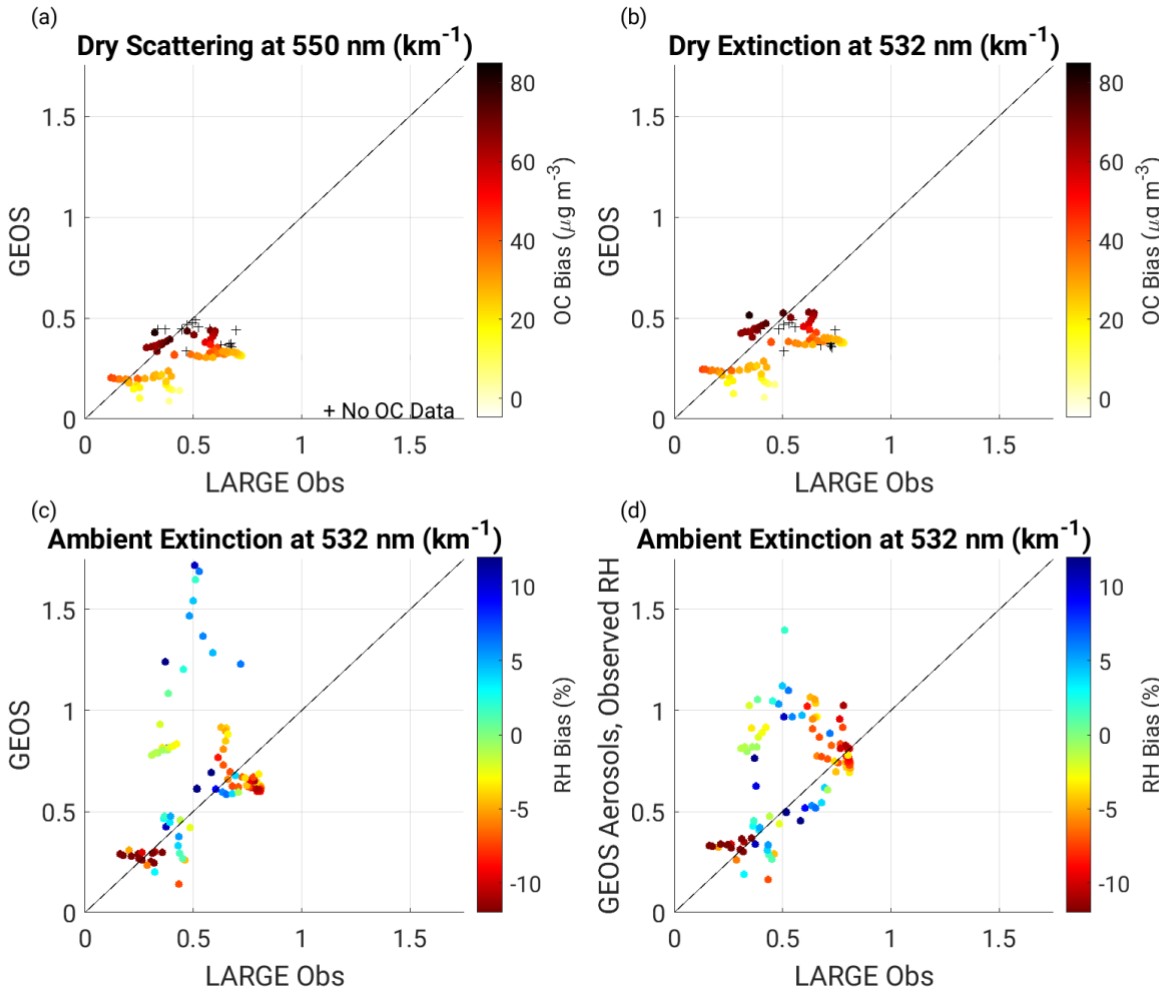

**Figure 10. Scatter plot of observations from LARGE versus GEOS GOCART2G for (a) dry scattering at 550 nm, (b) dry extinction at 532 nm, (c) ambient extinction at 532 nm, and (d) ambient extinction at 532 nm with bias corrected relative humidity in GEOS. Panels (a) and (b) are coloured based on the bias in organic carbon mass concentration and panels (c) and (d) are coloured based on the bias in relative humidity.**





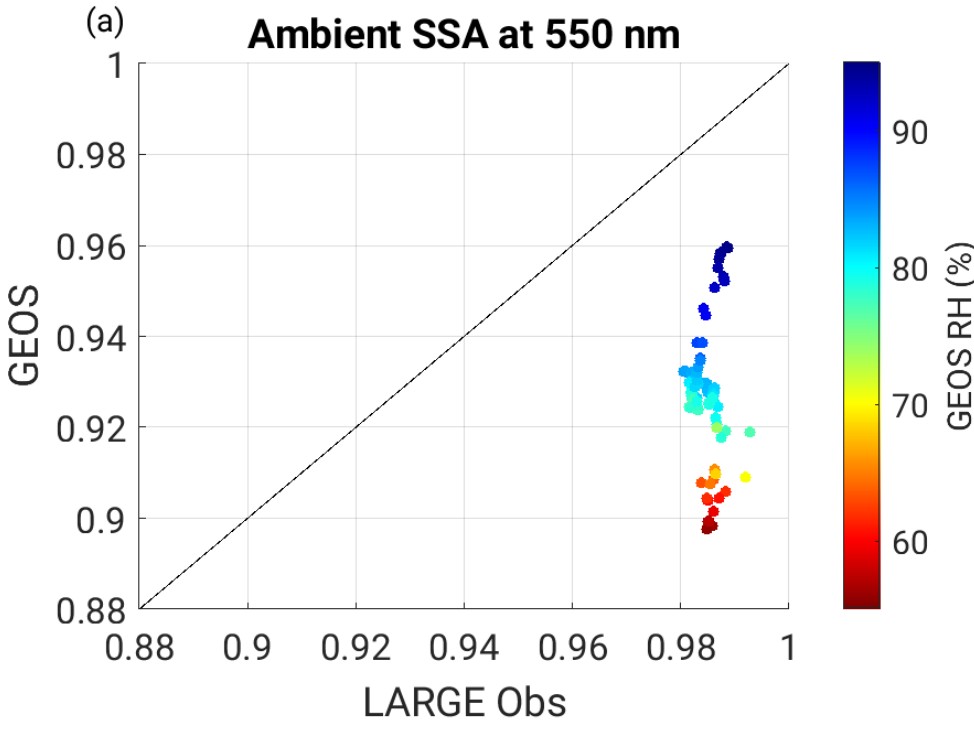

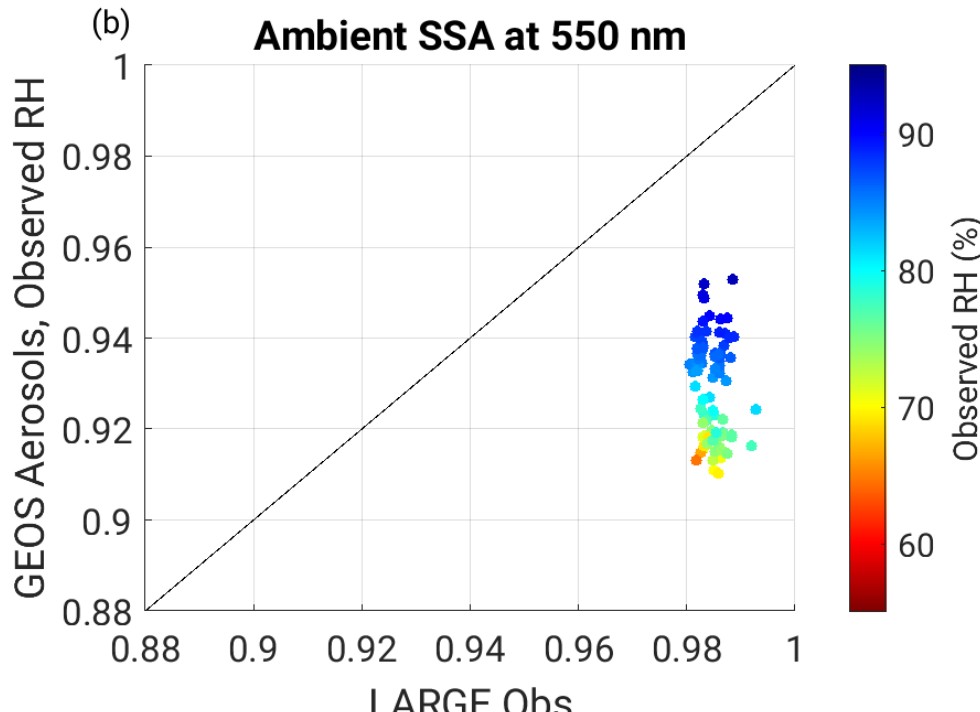





**Figure 11. Ambient single scattering albedo at 550 nm computed using (a) the relative humidity from GEOS and (b) observed relative humidity from the aircraft versus the observed single scattering albedo for the flight segment from RF9. Points are coloured based on the relative humidity in GEOS in (a) and the observed relative humidity in (b).**

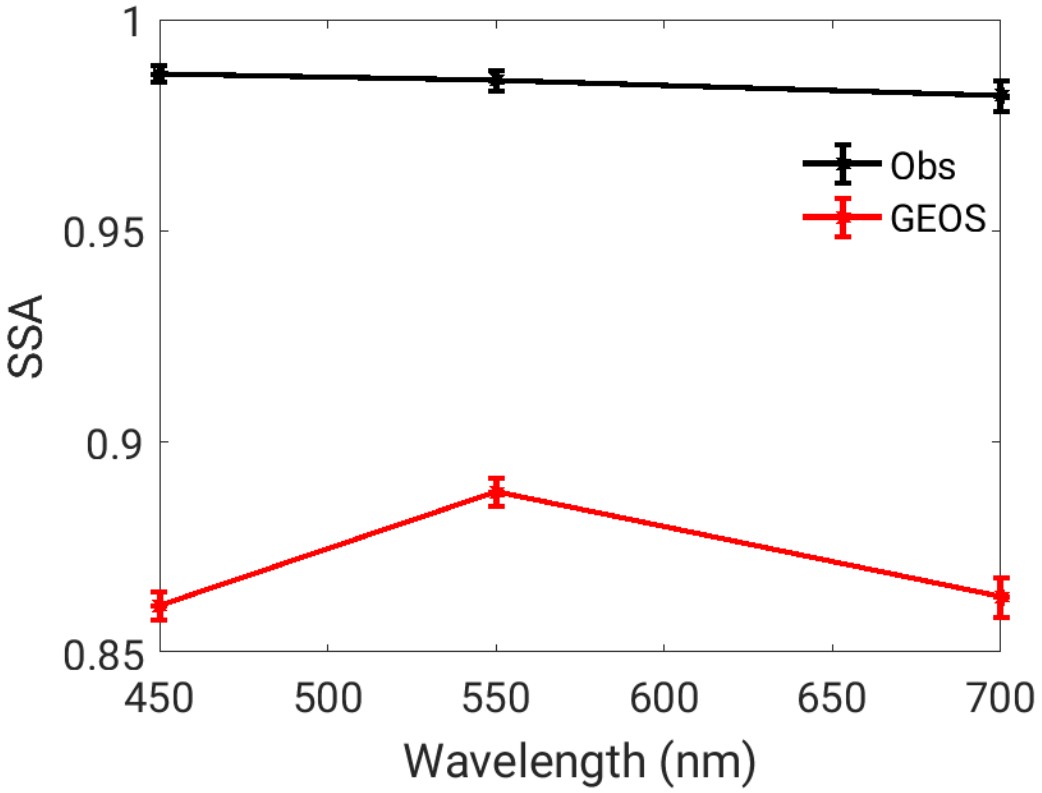

**Figure 12. Mean dry single scattering albedo at 450 nm, 550 nm, and 700 nm from GEOS and the LARGE observations for the flight**
**transect from RF9. Error bars indicate one standard deviation from the mean.**





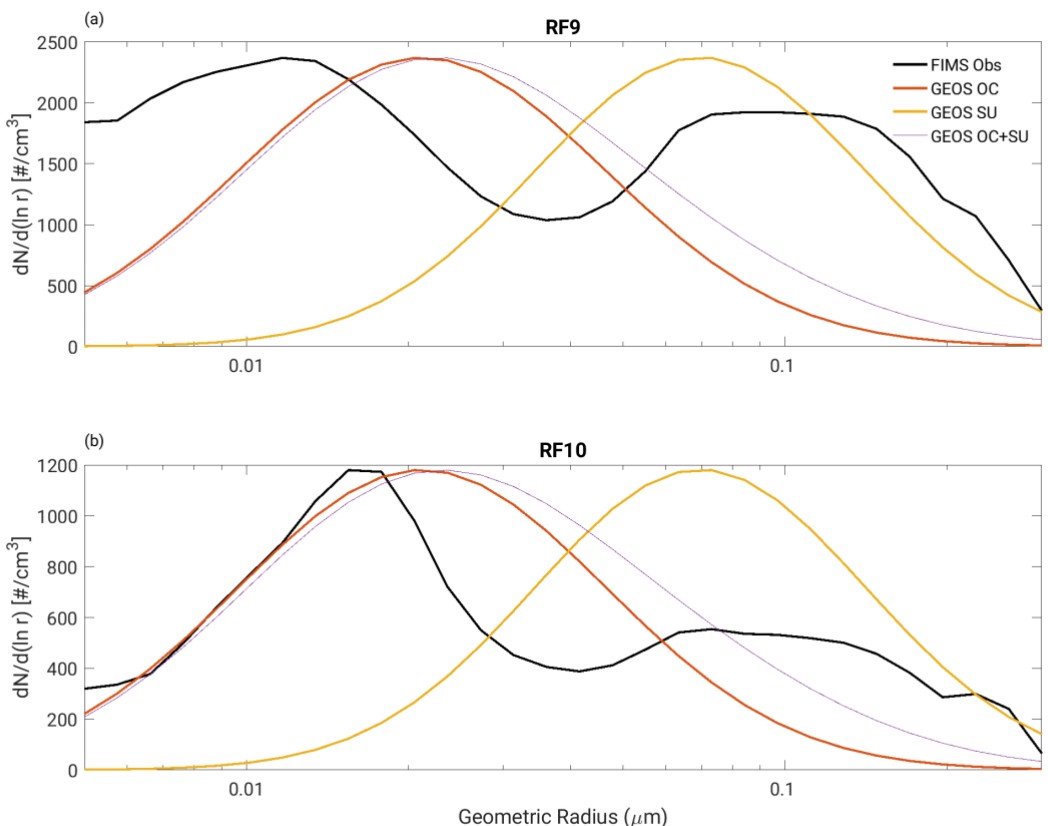

**Figure 13.** The observed dry aerosol size distribution from FIMS for data points classified as a biomass burning regime from (a)
RF9 and (b) RF10. Also shown is the assumed particle size distribution for organic carbon and sulphate in GEOS, scaled to match
the peak in the observed distributions, as well as those distributions linearly added for a more direct comparison between GEOS
and FIMS.




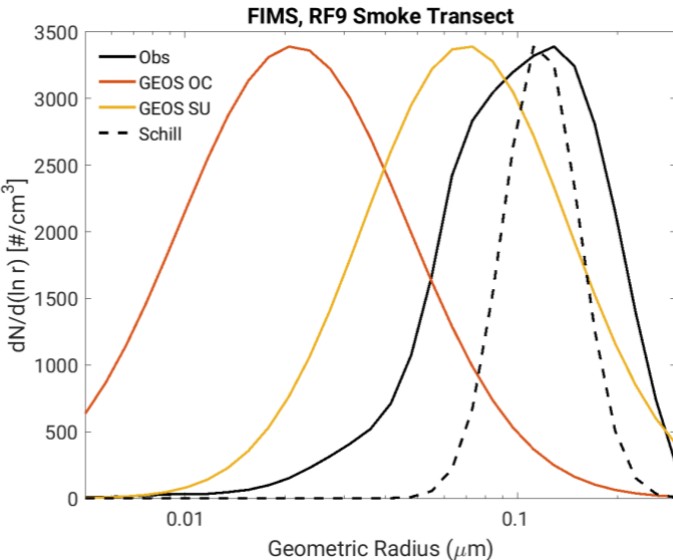

**Figure 14. The observed dry aerosol size distribution from FIMS for the flight segment from RF 9 as well as the assumed particle size distributions for organic carbon and sulphate in GEOS and the Schill size distribution derived from data collected during SEAC4RS (Schill et al., 2021).**