# Peer review of "An Evaluation of Biomass Burning Aerosol Mass, Extinction, and Size Distribution in GEOS using Observations from CAMP2Ex"

_Atmospheric Chemistry and Physics, 2022_

## Author Comment (AC1)

This manuscript presents a thorough evaluation of the performance of the GEOS global atmospheric chemistry model with the GOCART aerosol parameterization. The model output is compared with in situ, airborne measurements of aerosol compositional, microphysical, and optical properties made in the vicinity of the Philippines as part of the CAMP2EX project in 2019.

The output from the model is compared with the observations over project-wide averages, and for some specific flights in biomass burning smoke, and includes both extensive and some intensive parameters. The importance of assimilation of AOD values is examined, as are the effects of a newer convective parameterization scheme. While not comprehensive, this analysis is enough to identify discrepancies in the assumptions regarding aerosol composition, hygroscopicity, and size distribution in the GOCART parameterization. These discrepancies are large enough to be a concern, and indicate that perhaps the GOCART approach, with its one-size-fits-all static parameterization of the properties of different aerosol components, should be replaced with a more physically based and interactive aerosol parameterization.

This manuscript is suitable for publication in ACP following revisions that fall somewhere between major and minor (I've described them as minor). I have one significant comment below that I would definitely like to see addressed prior to publication, followed by a number of minor comments. Despite the number of comments, this is a generally well written paper, and an interesting and enjoyable one to read.

Thank you for taking the time to read our manuscript and provide feedback! Please find our responses to the comments written in red below. Of most importance, a new figure has been added to the manuscript to further investigate scattering at 80% relative humidity and the extrapolation that is performed in the observations to arrive at ambient scattering under humid conditions.

Major Comment:

Much of the analysis relies on comparison of the model output with extinction values at ambient RH. These are referred to as "measurements", but in fact they are derived values that rely implicitly on an underlying assumption regarding aerosol hygroscopicity. As documented in the headers of the data files from the LARGE measurements for this campaign, scattering (which vastly dominates extinction here) was measured at two fixed RH values: ~20% and ~80% RH. The ambient scattering is then calculated by assuming a functional form (the gamma parameterization), which is just a power law fit. In other words, the ambient scattering (hence extinction) is based on fitting a parameter to the two measurement points, then extrapolating to the ambient RH. The error in this approach is probably not large for RH values lying between or close to the two measurement RH values (20% and 80%). However, in the CAMP2EX profiles, the ambient RH exceeds 90% in the upper half of the boundary layer, and this is where the greatest contribution to extinction lies. In this high-RH region, the power-law hygroscopic growth curve pitches very sharply upwards, and small errors in the assumed shape of this growth curve can amplify to very large errors in calculated ambient extinction. So this comparison is not optimal, because the ambient extinction values will have large, and unknown, uncertainties in the high-RH region that dominates AOD.

One of the goals of the comparison of the measured and modeled extinction is to determine if the model hygroscopic growth is consistent with the measurements. To accomplish this, it might make more sense to compare the model to the measurements for the actually measured extinction values at the 20% ("dry") and 80% ("wet") conditions. That way the comparison is not between the modeled values

and those derived from the measurements with an assumed shape to the hygroscopic growth curve. Alternatively, you could plot the full f(RH) curve the model would produce (for a given location and time), from 20% RH to ~95% RH, and compare that with the same curve provided by the gamma parameter calculated from the LARGE measurements. This would be informative, and would allow you to compare the extinction at the measured 20% and 80% RH values as well as look at the response at higher RH values. The ratio of these two curves would indicate where there might be relative biases (although whether these lie in the model or in the gamma parameterization assumption might not be so clear).

Thank you for pointing out the extrapolation for scattering above 80% RH, which is indeed the case. The contribution of absorption to the total ambient extinction does not account for hygroscopic growth as indicated by the header in the data files. As suggested, a new figure has been added to the manuscript (and copied below) that shows scattering at 80% from the observations and the model.  The top panel demonstrates that there is decent agreement in scattering between LARGE and GOCART2G when the overestimation of organic carbon is not exceptionally large. Otherwise, there are data points that indicate an overestimation in scattering, somewhat like the overestimation in extinction when using the bias corrected relative humidity shown in Figure 10. The bottom panel illustrates that there is a stronger relationship between scattering and relative humidity in the model when the observed relative humidity is above 80%. Given the extrapolation for the observations, it is difficult to determine whether the model or observations are responsible for all or some of the bias.

[Figure]

Minor Comments:

There are a few places where there are typos or where the manuscript could be edited for clarity.

1) Lines 81-85. The FIMS and HSRL2 are introduced by name, but the text doesn't say what they measure. That information appears in lines 100-104. I was trying to figure out what a FIMS measured throughout the text between these lines. Suggest moving the latter section up to lines 81-85.

Clarification has been added to the line in question, which now reads as "Here, we make use of the NASA Langley Aerosol Research Group Experiment (LARGE) suite of instruments, particle size distribution from the Fast Integrated Mobility Spectrometer (FIMS; Kulkarni and Wang, 2006; Wang et

al., 2017), and aerosol scattering profiles from the 2nd generation High Spectral Resolution Lidar (HSRL2) (Burton et al., 2018) as summarized in Table 1."

2) Line 117. Change "hydroscopic" to "hygroscopic".

This should have been hydrophobic and has been corrected.

3) Line 123. Define sigma as the geometric standard deviation.

This has been updated.

4) Line 130. Change to, "The underlying meteorology from GEOS is used for horizontal and vertical transport and deposition of all of the aerosol species, as well as wind-driven emissions of dust and sea salt."

This has been changed as suggested.

5) Line 193. Define "lidar ratio".

Lidar ratio is now defined.

6) Lines 211 and 223. "Optical array" has a particular meaning in optical design. Suggest changing to "optical properties instruments" or something similar.

Array has been changed to instrument suite.

7) Line 213. Change to "fine particles that are efficiently sampled".

Fixed.

8) Lines 222-234. Here is where you could expand on the model/measurement comparison of extinction as a function of RH and focus on the actual full frh curve or compare the extinctions at 20% and 80% RH.

Thank you for the suggestion. We opted to discuss the scattering at 80% RH in Section 3.2 since that is where f(RH) is discussed.

9) Line 241. I don't understand this sentence. "Analysis increment"?

Analysis increment quantifies the adjustment in aerosol mass that occurs in response to the assimilation of aerosol optical depth. This sentence has been modified to "The assimilation of aerosol optical depth results in an increase of black carbon mass and a doubling of the positive bias in the mass concentration within the boundary layer."

10) Line 258. Please add a comma between "region" and "or".

Done.

11) Line 273. Please change to, ". . .represented that percentage of organic carbon well; however, it struggled with. . . ."

Fixed.

12) Line 281. In the heading for Table 2, I believe this in not "total aerosol mass" because it excludes BC.

Yes, "total" has been removed from the heading for Table 2 (now Table 3).

13) Line 299. I believe the values for the extinction measurements are 20% and 80% RH, at least according to the file headers.

Yes, correct. Gamma is found using 40% however f(RH) is determined by using gamma at 20% and 80%, which is now noted in the text.

14) Line 313. Saying the GEOS values for SSA are "below 0.96" doesn't do justice to the magnitude of the discrepancy, which is quite large. Maybe say they range from "0.9 to 0.96". The co-albedos disagree by a factor of 2-4; this is quite large in the context of direct radiative effects.

This sentence has been updated to "Nearly all points for the observations have a SSA greater than 0.98 while the SSA in GEOS ranges from ~0.9 to ~0.96".

15) Line 323-324. It's true that the extinction could be juiced up by making the modal diameter larger and the width wider, but the assumed standard deviation of ~2+ is already quite a bit larger than literature values would support. You might want to lead into the next sentence by saying, "We will examine the in situ measurements to see if such changes could be justified. We begin by looking at . . . ."

This sentence has been modified to "We begin by looking at two flights, RF9, the flight examined in Section 2.2, and RF10, which also captured aged biomass burning aerosol but downstream in the Philippine Sea on the following day, to determine if changes to the particle size distribution are justified."

16) Line 337. The sentence beginning "The primary peak" is confusing. Please define what you mean by the "primary peak? It's being "shifted toward a larger radius" than what? Are these the measured or GEOS size distributions you're talking about? Please clarify.

This sentence has been modified to "The primary peak in the size distribution, centered at ~0.015 μm in the observations, is shifted towards a larger radius in GEOS."

17) Line 368. I'm not sure what "mass piling up at the top of the PBL" means. I don't think mass (of air or of aerosol) can "pile up", at least not without increasing air density a lot!

This sentence has been updated to "Using aerosol mass concentration as a tracer indicates an overestimate of aerosol within the boundary layer in GEOS, particularly at the top of the PBL such that the aerosol is unable to sufficiently penetrate into the free troposphere."

18) Lines 391-395. These sentences are unclear. What "concern" about the size distribution could be "rectified"? Please be specific, e.g., "agreement between modeled and measured extinction at high RH could be improved if the modeled size distribution had a larger modal radius and/or a larger standard deviation. However, these adjustments are not supported by the FIMS size distribution measurements. In fact, the observed mode radius is in excellent agreement. . . ."

This section of the conclusions has been re-written as copied below:

"Agreement between the modelled and observed extinction could be improved by adjusting the assumed particle size distribution for brown carbon to have a larger mode radius or a wider distribution. A comparison of the assumed particle size distribution to observations from FIMS indicated that the

current mode radius is too small. The observed mode radius is in excellent agreement with observations from past field campaigns, however the modal width for CAMP²Ex is larger. A limitation of GOCART, however, is that the particle size distribution cannot vary in time or space for a given aerosol species. This contradicts the variability seen within particle size distribution during the low-level flight segment from CAMP²Ex's RF9 as well results from other field campaigns that indicate the particle size distribution of biomass burning aerosol changes with respect to median diameter and modal width as smoke ages (June et al., 2022).  If the assumed particle size distribution for brown carbon in GEOS were to be modified such that the mode radius is larger, it would need to be thoroughly evaluated as a change in the particle distribution would also impact fields such as the single scattering albedo and Angstrom exponent. Ultimately, a more physically based aerosol module would need to be used for GEOS to accurately represent the variability in the particle size distribution."

19) References. Please ensure that all references are compliant with Copernicus formatting guidelines. For example, some of the references (e.g., Burton et al., 2012) have capitalized titles, while most do not. This is a result of reference management software, which always needs to be thoroughly checked manually.

The citations have been standardized such that titles are no longer capitalized.

20) References. I'm not sure Schill et al. is citable--it's an unpublished conference presentation.

A section has been added to the supplemental material to further describe the particle size distribution developed by Schill et al. in lieu of the citation, which has been copied below.

Merging the Particle Analysis by Laser Mass Spectrometry (PALMS) number fractions with an independently measured, quantitative size distribution to make species-specific (e.g., biomass burning, dust, sea salt) size distributions has been detailed in several publications (e.g., Froyd et al., 2019; Brock et al., 2021). The biomass-burning-only size distributions from the Studies of Emissions and Atmospheric Composition, Clouds and Climate Coupling by Regional Surveys (Toon et al., 2016) and NASA Atmospheric Tomography (Thompson et al., 2022) missions are used. Using both missions allows us to assess the biomass-burning-only size distributions from plumes <1 day old to biomass burning aerosol that have aged in the atmosphere ~30 days. We find that, regardless of age, the biomass burning size distribution can be described by a single lognormal mode. In this work, we use the size distributions from fires <5 days old. The $r_n$, median and $\sigma$ values from these size distributions are 0.1175 μm and 1.3, respectively.

Brock, C. A., Froyd, K. D., Dollner, M., Williamson, C. J., Schill, G., Murphy, D. M., . . . Wofsy, S. C. (2021). Ambient aerosol properties in the remote atmosphere from global-scale in situ measurements. Atmospheric Chemistry and Physics, 21 (19), 15023–15063. doi: 10.5194/acp-21-15023-2021.

Froyd, K. D., Murphy, D. M., Brock, C. A., Campuzano-Jost, P., Dibb, J. E., Jimenez, J.-L., . . . Ziemba, L. D. (2019). A new method to quantify mineral dust and other aerosol species from aircraft platforms using single-particle mass spectrometry. Atmospheric Measurement Techniques, 12 (11), 6209–6239. doi: 10.5194/amt-12-6209-2019.

Thompson, C. R., Wofsy, S. C., Prather, M. J., Newman, P. A., Hanisco, T. F., Ryerson, T. B., . . . Zeng, L. (2022). The NASA Atmospheric Tomography (ATom) Mission: Imaging the Chemistry of the Global

Atmosphere. Bulletin of the American Meteorological Society, 103 (3), E761–E790. doi: 10.1175/BAMS-D -20-0315.1.

Toon, O. B., Maring, H., Dibb, J., Ferrare, R., Jacob, D. J., Jensen, E. J., . . . Pszenny, A. (2016). Planning, implementation, and scientific goals of the Studies of Emissions and Atmospheric Composition, Clouds and Climate Coupling by Regional Surveys (SEAC 4 RS) field mission. Journal of Geophysical Research: Atmospheres, 121 (9), 4967–5009. doi: 10.1002/2015JD024297

21) Figures. The figures are generally very nicely done, clearly labeled. However, I'm not sure of the ability of those with color impairment to read the shaded vertical profiles (e.g., Figs. 3-6). Also, in these figures the horizontal lines indicating HSRL2 MLH and GOES PBLH are difficult to discern; the black and blue colors are quite close. Could one line be made dotted? In Fig. 4 you might need a 3rd line type for clarity.

The figures were uploaded to https://www.color-blindness.com/coblis-color-blindness-simulator/ and checked for clarity for those with a color impairment. Modeled PBL height now uses a dotted line as shown below.

[Figure]

22) Figures 13, 14. It might be nice to fit a lognormal to the FIMS peaks, then you could directly compare the mode diameter and standard deviation with the model. My eye says that the standard deviation for

the measurements is less than the very broad modeled values, but I can't be sure without fitting. I much prefer the aspect ratio of Fig. 14 to that of Fig. 13; could Fig. 13 be made with side-by-side plots, rather than vertically stacked ones? This would make Fig. 13 and 14 look more similar.

The sub-flight variability and averaging make it difficult to accurately fit a lognormal distribution to the observations in Figure 13, but a lognormal fit was performed for the flight segment shown in Figure 14. The low-level segment through the smoke plume indicated a median radius of 0.0995 µm and a modal width of 1.77, which would be a narrower distribution with a larger radius than what is assumed by GEOS. The aspect ratio of Figure 13 was modified and is shown below.

[Figure]

---

## Author Comment (AC2)

The authors evaluate the effects of recent modifications in the GEOS model (version 5.22, 5.25 and GOCART2G) on aerosol-related estimations. To that effect, they use remote sensing and in situ measurements from the recent CAMP²EX airborne field campaign (Philippines, Aug-Oct 2019). Their study focuses on the evaluation of modeled Biomass Burning (BB) aerosol speciated mass and total backscatter, scatter, extinction, single scattering albedo and size distribution as well as modeled relative humidity, temperature, and planetary boundary layer height. This paper is well structured, its results important but needs clarification in many places. It will be worthy of publication once the issues below are properly addressed.

Thank you for providing a detailed review of the manuscript. The clarifications requested have improved readability and reproducibility.

**Major comments:**

. We recommend the authors combine a few or move some figures to the appendix, especially the ones that are barely analyzed in the text (e.g., Fig. 9).

Figure 9 has been moved to the supplemental document.

. The recent presence of nitrate and Brown Carbon (BrC) aerosols in GOCART and GOCART2G needs more references and descriptions. We recommend that the authors include a table of microphysical and optical properties for all the species present in the model.

Tables detailing the size bins and optics look up tables have been added to the supplemental material and are copied below.

**Table S1. The radii and densities of the five size bins for dust used for settling in GEOS.**

|                        | DU001 | DU002 | DU003 | DU004 | DU005 |
|------------------------|-------|-------|-------|-------|-------|
| Radius (µm)            | 0.73  | 1.4   | 2.4   | 4.5   | 8     |
| Radius Lower Bound (µm)| 0.1   | 1     | 1.8   | 3     | 6     |
| Radius Upper Bound (µm)| 1     | 1.8   | 3     | 6     | 10    |
| Density (kg m$^{-3}$)  | 2500  | 2650  | 2650  | 2650  | 2650  |

**Table S2. The radii and densities of the five size bins for sea salt used for settling in GEOS**

|                        | SS001 | SS002 | SS003 | SS004 | SS005 |
|------------------------|-------|-------|-------|-------|-------|
| Radius (µm)            | 0.079 | 0.316 | 1.119 | 2.818 | 7.772 |
| Radius Lower Bound (µm)| 0.03  | 0.1   | 0.5   | 1.5   | 5     |
| Radius Upper Bound (µm)| 0.1   | 0.5   | 1.5   | 5     | 10    |
| Density (kg m$^{-3}$)  | 2200  | 2200  | 2200  | 2200  | 2200  |

**Table S3. The radii and densities used for the settling of ammonium and nitrate in GEOS. Note that NI002 and NI003 represent coarse mode nitrate formed from the heterogenous production of nitrate from sea salt and dust, respectively.**

|  | NH$_4^+$ | NI001 | NI002 | NI003 |
|---|---|---|---|---|
| Radius (µm) | 0.2695 | 0.2695 | 2.1 | 7.57 |
| Density (kg m$^{-3}$) | 1769 | 1725 | 2200 | 2650 |

**Table S4. File versions for the optics look up tables used for GOCART2G. These files are available for download at https://portal.nccs.nasa.gov/datashare/iesa/aerosol/AerosolOptics/.**

| Aerosol Species | Optics File |
|---|---|
| Black Carbon | opticsBands_BC.v1_3.RRTMG.nc |
| Brown Carbon | opticsBands_BRC.v1_5.RRTMG.nc |
| Organic Carbon | opticsBands_OC.v1_3.RRTMG.nc |
| Dust | opticsBands_DU.v15_3.RRTMG.nc |
| Sea Salt | opticsBands_SS.v3_3.RRTMG.nc |
| Nitrate | opticsBands_NI.v2_5.RRTMG.nc |
| Sulfate | opticsBands_SU.v1_3.RRTMG.nc |

. The changes applied to the different models in Table 1 should be further described and the authors should focus on explaining the potential effects of these changes on the modeled aerosol microphysics, spatial distribution, optical properties etc.

The sentence below has been added to Section 2.2 to indicate the impact of model changes on aerosols.

"The changes in convection have the potential to alter the vertical transport of aerosols as well as relative humidity, which is passed to the optics look up table to determine aerosol scattering and extinction."

. A high-level diagram illustrating the different modules in the model as well as the many changes in Table 1 would be helpful. The diagram could also emphasize what this paper has investigated in more detail (e.g., RH).

GEOS contains multiple modules, each with underlying components, to represent the Earth system such that a description of such a diagram would be too extensive for the main body of the text. Instead, diagrams have been added to the supplemental material and are copied below for reference. Changes that were implemented from the baseline, GEOS 5.22, have been highlighted in red.

[Figure]

(a)

[Figure]

(b)

[Figure]

(c)

. Throughout the paper, we recommend a clear discussion of all the error sources in the model.

Several items have been clarified as a result of the detailed comments, which we hope elucidates the error sources in the model.

**Detailed Comments:**

. Line 14: "serving as cloud condensation nuclei". As written, it seems that this is the only way BB impacts radiative forcing. We recommend either re-wording or adding direct and semi direct radiative effects as well.

This sentence now states " Biomass burning aerosol impacts aspects of the atmosphere and Earth system through the direct and semi-direct effects, as well as influencing air quality.

. Line 19: The authors should be clearer on which satellite/ ground-based sensor(s) is(are) used and in which model version.

This has been clarified to state that MODIS and Aeronet are assimilated in the operational configuration of GEOS.

. Line 24: Why not say "Aerosol extinction within GEOS is a function of the mass of different aerosol species, the ambient relative humidity, the assumed spectral optical properties and particle size distribution per species".

This sentence has been updated as suggested.

. Line 25: "aggressive" is not usually used in that case. Maybe "high" or "overestimated".

"High" is now used.

. Line 27: "a mode radius" – does GEOS assume only one size mode for its particle size distribution of OC? Aerosols are usually (at least) bi-modal so this should be discussed.

Yes, only one size mode is used for OC and this is now specified. The use of a single lognormal distribution for each aerosol species stems from the OPAC database described in section 3c of Hess et al. (1998). A citation for Hess et al. (1998) has been added to Section 2.2 where the model is described.

Hess, M., Koepke, P., & Schult, I. (1998). Optical Properties of Aerosols and Clouds: The Software Package OPAC, Bulletin of the American Meteorological Society, 79(5), 831-844.

. Line 31: See comment for line 14.

This sentence now states "Aerosols are an important component of the Earth system due to their role in the direct and semi-direct effects and impact on air quality."

. Line 37: "smoke and biomass burning aerosol" reads as if these two things were different.

"Smoke" has been removed.

. Line 49: SSA and its link to aerosol light absorption needs to be briefly described; and we recommend writing "… due to different assumptions for aerosol…".

SSA has been defined.

. Line 51: We recommend simplifying and writing "An additional source of uncertainty would be the biomass burning aerosol emissions".

This sentence has been simplified as suggested.

. Line 59: This should be "anthropogenic". The distinction between "white (anthropogenic) and brown (biomass burning, BB) OC" is not clear. Some BB aerosols can be labelled anthropogenic (e.g., prescribed fires) and it's not clear what the authors mean by "white" OC. This needs more description.

We have tried to make this clearer in the text by specifying organic carbon from fires versus other sources.

. Line 66: "two moment cloud microphysics" is mentioned abruptly here, with no obvious link to what was written previously or afterwards. If kept in the text, "two moment cloud microphysics" should also be described.

"Clouds" has been added to the previous sentence.

. Line 73: "future" is written twice.

Thanks, this has been adjusted.

. Line 80: "over the Philippine Sea".

"The" has been added.

. Line 82: A table listing the instruments, measurements, size ranges, resolutions and references is recommended here, like Table 1 in Edwards et al., [2021].

In an effort to not duplicate the table in Edwards et al (2021) since the instrumentation used is nearly identical, we opted to add a table of the measurements with their uncertainty and cite Edwards et al. (2021) for additional details.

. Line 85: Is the AMS instrument operated by the LARGE team during CAMP²EX? It usually isn't the case so that would be new. It does not seem to be the case in Stahl et al. [2021]. Please check and clarify.

Yes, the AMS was operated by LARGE with Luke Ziemba as the PI.

. Line 92: "50% uncertainty" needs a reference or "[personal communication from …]" and if true, this should be better explained. Again, it was not mentioned in Stahl et al. [2021].

The uncertainty is listed in the ict data files with the header information. The following text was added to specify where this came from:

"Uncertainty for AMS-derived mass concentrations is driven by variability in the instrument collection efficiency (CE), which is a scalar term with typical values from 0.5 to 1.0 for the standard conical tungsten vaporizer, depending on particle composition and phase (Hu et al., 2018).  For CAMP2Ex analysis, mass concentrations are derived using a constant value of 1.0 based on comparison with independent measurements from a particle-into-liquid sampler (PILS).  Still, a conservative value of 50% uncertainty is used to account for the unknown CE and is generally consistent with other aircraft AMS measurements (Bahreini et al., 2009)."

Hu et al.: https://pubs.acs.org/doi/full/10.1021/acsearthspacechem.8b00002?casa_token=pNt0vw1-nFcAAAAA%3AqJZSBKVfadOA6MVdBZ7ji0QWKmWHCxYQaqreq-Df8aOBBj3sLdPcIylqJbBv_KMsEjgqTLKh5Sk2XjSM

Bahreini et al.: https://agupubs.onlinelibrary.wiley.com/doi/epdf/10.1029/2008JD011493

. Line 93: "inconsistencies between measured mass concentrations and optical properties". This is not clear. It should be explained/ illustrated/referenced.

This is an ongoing topic of discussion among the CAMP2Ex science team that has included co-authors here. Unfortunately, details have not been published yet and it is beyond the scope of this paper. The primary message is that the dataset is not ideal for computing mass extinction efficiency in its current form.

. Line 117: GOCART (legacy) usually uses [e.g., Chin et al., 2002, 2009, 2014] as references. Also, GOCART was already introduced in line 62 (with the right reference i.e., Chin et al., 2002).

An additional reference for Chin et al. (2004) has been added for GOCART. Colarco et al. (2010) is the appropriate reference for the online coupling of GOCART to GEOS.

. Line 117: should read "hydrophilic and … hydrophobic" or "hygroscopic and… hydrophobic".

Correct! This was also caught by reviewer 1 as well! The sentence now states "hydrophilic" and "hydrophobic".

. line 118: "size bins per model species" is not clear here. It should be added that nitrate was not originally included in GOCART [e.g., Chin et al., 2002, 2009, 2014] but was developed more recently in the GMI model [Bian et al., 2017], which is an off-line chemistry model. It was then later implemented in the GEOS/GOCART model, followed by GOCART2G. As for BrC, the authors should specify that is it present in both GOCART and GOCART2G? If there is a specific reference describing GOCART2G, the authors should mention it. If not (Colarco et al., 2017 might be the only "indirect" reference), they should describe what is meant by BrC chemically and optically.

A publication detailing GOCART2G is not available at the present time. Brown carbon is not included operationally in legacy GOCART and was added as part of GOCART2G. Prior to GOCART2G, organic carbon was emitted in the model using two sources of data that were combined and treated as a single aerosol tracer termed organic carbon: 1) the Quick Fire Emissions Dataset (QFED) and 2) from a global emissions dataset such at CEDS or HTAP/EDGAR, that could be further divided into emissions from energy production, transportation, etc. If a fire is present in QFED, it does not matter if it is prescribed or induced anthropogenically. Upon the implementation of GOCART2G, the brown carbon tracer only gets emissions from QFED while organic carbon only gets emissions from CEDS. Differences with respect to optics can be seen by comparing the optics tables as specified in opticsBands_BRC.v1_5.RRTMG.nc and opticsBands_OC.v1_3.RRTMG.nc, available for download at https://portal.nccs.nasa.gov/datashare/iesa/aerosol/AerosolOptics/.

A reference to Bian et al. (2017) was added for nitrate as well as a sentence pertaining to brown carbon. The paragraph had a lot of changes, and the new text has been copied below for reference.

"Within GEOS, aerosols are governed by the GOCART module (Chin et al., 2002; Chin et al., 2004; Colarco et al., 2010). This module simulates the transport and optical properties of externally mixed hydrophobic and hydrophilic organic and black carbon, sulphate, three size bins for nitrate (implemented in the same manner as Bian et al. (2017)), five size bins for sea salt, and five size bins for dust. To implement updates and allow for future development, the (legacy) GOGART module code had been refactored and termed "GOCART2G". GOCART2G now includes brown carbon as a new radiatively interactive species. Following Colarco et al. (2017), biomass burning emissions of organic aerosol are assigned to the brown carbon species, while other anthropogenic and biogenic sources are assigned to the legacy organic carbon tracer. A new mechanism secondary production of both brown and organic carbon is adopted based on oxidation of volatile organic carbon (VOCs) scaled to carbon monoxide emissions following Hodzic and Jimenez (2011). Brown carbon is treated chemically the same as organic carbon in GOCART2G but is assigned optical properties that have spectrally varying absorption in the shortwave, consistent with observations from the space-based Ozone Monitoring Instrument (Colarco et al. 2017). Other aerosol species optical properties are primarily based on the Optical Properties of Aerosols and Clouds (OPAC) database described by Hess et al. (1998), except dust which is based on Colarco et al. (2014). Details pertaining to the optics look up tables can be found in Table S4 in the supplemental material. Sulphate, black carbon, brown carbon, and organic carbon are assumed to have a lognormal size distribution with number mode radii for dry particles of 0.0695 µm, 0.0188 µm, 0.0212 µm, and 0.0212 µm, respectively and a geometric standard deviation of 2.03, 2, 2.2, and 2.2 respectively."

. Line 121: "The optics look up tables for each aerosol species are the same as described by Colarco et al. (2017)". We can't seem to find these look up tables in Colarco et al. (2017). We recommend that this paper adds a table describing these optical properties.

Tables detailing the size bins and optics for each aerosol species have been added to the supplemental material and are copied below for reference.

**Table S1. The radii and densities of the five size bins for dust used for settling in GEOS.**

|  | DU001 | DU002 | DU003 | DU004 | DU005 |
|---|---|---|---|---|---|
| Radius (µm) | 0.73 | 1.4 | 2.4 | 4.5 | 8 |
| Radius Lower Bound (µm) | 0.1 | 1 | 1.8 | 3 | 6 |
| Radius Upper Bound (µm) | 1 | 1.8 | 3 | 6 | 10 |
| Density (kg m$^{-3}$) | 2500 | 2650 | 2650 | 2650 | 2650 |

**Table S2. The radii and densities of the five size bins for sea salt used for settling in GEOS**

|  | SS001 | SS002 | SS003 | SS004 | SS005 |
|---|---|---|---|---|---|
| Radius (µm) | 0.079 | 0.316 | 1.119 | 2.818 | 7.772 |

| | 0.03 | 0.1 | 0.5 | 1.5 | 5 |
|---|---|---|---|---|---|
| Radius Lower Bound (µm) | 0.03 | 0.1 | 0.5 | 1.5 | 5 |
| Radius Upper Bound (µm) | 0.1 | 0.5 | 1.5 | 5 | 10 |
| Density (kg m$^{-3}$) | 2200 | 2200 | 2200 | 2200 | 2200 |

**Table S3. The radii and densities used for the settling of nitrate and ammonium in GEOS. Note that NI002 and NI003 represent coarse mode nitrate formed from the heterogenous production of nitrate from sea salt and dust, respectively.**

| | NH$_4^+$ | NI001 | NI002 | NI003 |
|---|---|---|---|---|
| Radius (µm) | 0.2695 | 0.2695 | 2.1 | 7.57 |
| Density (kg m$^{-3}$) | 1769 | 1725 | 2200 | 2650 |

**Table S4. File versions for the optics look up tables used for GOCART2G. These files are available for download at https://portal.nccs.nasa.gov/datashare/iesa/aerosol/AerosolOptics/.**

| Aerosol Species | Optics File |
|---|---|
| Black Carbon | opticsBands_BC.v1_3.RRTMG.nc |
| Brown Carbon | opticsBands_BRC.v1_5.RRTMG.nc |
| Organic Carbon | opticsBands_OC.v1_3.RRTMG.nc |
| Dust | opticsBands_DU.v15_3.RRTMG.nc |
| Sea Salt | opticsBands_SS.v3_3.RRTMG.nc |
| Nitrate | opticsBands_NI.v2_5.RRTMG.nc |
| Sulfate | opticsBands_SU.v1_3.RRTMG.nc |

. Line 125: it would be better to write "bias corrected AOD observations from the Moderate Resolution Imaging Spectroradiometer (MODIS) aboard Terra and Aqua are assimilated in all the models of Table 1 except "No GAAS". Is MODIS the only sensor that is assimilated in all models and is AERONET only assimilated in GEOS 5.22 and not the rest?

This sentence has been updated as suggested. Aeronet was not assimilated in the GOCART2G run so that we could use it as an independent data source for validation of that model version, which makes MODIS the only source of observations for AOD that is assimilated in all three versions. Given that the CAMP2Ex flights were predominantly over ocean, assimilating Aeronet in GOCART2G would have had little impact on the data presented in the manuscript.

. Line 131: '… as well as deposition and wind-driven emissions of dust and sea salt". Why not wind-driven emissions of BB or urban pollution?

Biomass burning and urban emissions are controlled by factors other than wind and would therefore not be accurately represented.

. Line 137: This sentence is not clear and should be re-written.

This has been rewritten to "The meteorology was constrained in two manners. GEOS 5.22 and GEOS 5.25 used an online data assimilation system (DAS) that ran at the same time as the general circulation model to produce an analysis. For the GOCART2G and No GAAS simulations, the analysis produced from a previous simulation was used to nudge the meteorology without the computational burden, often referred to as a "Replay"."

. Line 155: What is meant by "The diurnal evolution of … the lower troposphere"?

"Evolution" has been replaced by "cycle" to use more common terminology.

. Line 157: "Relative humidity was selected for this evaluation since it is used in the optics lookup tables for aerosols". We recommend specifying "in the model" here. This is where more information on the model and its different modules would be helpful in the introduction. Why not plot modeled PBLH and measured MLH on Figure 1.

"Modelled" aerosols are now specified.

While PBLH and MLH could be plotted on Figure 1, the radiosondes were launched from the ship while the HSRL2 was aboard the aircraft. The time and location of the observations would not be consistent within the figure.

. Line 165: The angstrom exponent and its link to aerosol size should be explained

Thank you for the suggestion however we do not feel it is appropriate to describe the relationship between the Angstrom exponent and aerosol size as it is solely being used to convert the Aeronet observations of AOD to 550 nm for comparison to the model.

. Line 169: "or localized urban emissions not in the CEDS emissions dataset" This is not clear. The first part of the sentence is about AERONET measurements (and sources of errors in the Level 1.5 data) and the second part seems to be about the model.

This sentence has been rephrased.

. Line 176: How are the MLH vs PBLH computed? Wouldn't we not expect MLH and PBLH to be the same? This paper is focused on evaluating modeled BB aerosol composition, microphysics, and optical properties. The authors should explain why they are also evaluating the simulated PBLH (e.g., impact on modeled aerosol vertical distribution).

MLH from the HSRL2 and PBLH from the model are not quite the same thing, which is why the same terminology is not used for both. Observed MLH is derived using aerosol backscatter as described by Scarino et al. (2014) while PBLH in GEOS is defined as the lowest model level in which the heat diffusivity falls below 2 $m^2s^{-1}$. Ideally, an instrument simulator would be used to find the MLH in GEOS using the same methodology as the HSRL2 however this tool has not been developed yet. Given that most aerosol mass is located within the boundary layer, PBL height is a useful quantity when analyzing the vertical profile of aerosols. This is now noted in the text.

. Line 177: Some figures show the three model versions of Table 1 and some do not. We recommend consistency. We recommend adding "(not shown here)" after "indistinguishable"

For consistency, we had placed the version of the figure with all three model version in the supplemental document. This is now indicated in the text.

. Line 179: "This trend … ".

This has been corrected.

. Line 182: We recommend describing the link between the spectral dependance of the aerosol backscatter and the size of particles.

The relationship between particle size and scattering at different wavelengths is now noted.

. Line 185: quantify "slight improvement"

This is quantified in Figure S2 within the supplemental document.

. Line 189: "there is a larger impact of the change in relative humidity between GEOS 5.22 and GEOS 5.25 than the aerosol updates implemented in GOCART2G" This sentence is not clear and illustrated.

This is demonstrated by the fact that the blue line for GEOS 5.25 is closer to the cyan line for GOCART2G than the red line for GEOS 5.22 in Figure 4b.

. Line 191: "compares well"

This has been fixed.

. Line 192: "is located too high due to the height of the boundary layer in the model." The PBL is defined as a strong gradient in the aerosol scattering profile. This feels like circular reasoning.

There are multiple definitions of the PBL height. While one can use the gradient of the aerosol scattering profile as a metric to determine the PBL height, it is the dynamics (or vertical profile of omega) that keeps the aerosol trapped within the boundary layer.

. Line 193: Why not show lidar ratios as well?

Thank you for the suggestion. Adding lidar ratios would be redundant as this information is already given in the included figures.

. Line 197: "based on the region of interest" makes it sound as if HSRL aerosol type is based on the location, but it is not.

This sentence has been rephrased to "Five aerosol types are considered here based on aerosol types typically present in the Philippines region: marine, polluted marine, smoke, fresh smoke, and urban pollution".

. Line 199: "the GEOS aerosol speciation for each HSRL2 derived aerosol type" – as the authors compare GEOS aerosol speciation for different aerosol types, why not analyze the results and evaluate whether the composition agrees well with the types in a quantitative way? Are the GEOS species "correctly translating" the HSRL aerosol types? This would be similar to the work of Kacenelenbogen et al. [2022]

Aerosol composition in GEOS associated with each HSRL2 derived aerosol type is included in the supplemental document as Figure S4.

. Line 200: "drastic decrease in the sample size above 2 km" should be quantified. And "There is also a focus placed on the GOCART2G" can be replaced by "We focus on…".

The number of observations for each aerosol type is given in the supplemental document. There are essentially no observations above 2 km. The following sentence has been modified as suggested.

. Line 202: Figure 5e – The authors should explain the differences between "fresh smoke" and "smoke" from HSRL

The difference between smoke and fresh smoke is described in Burton et al. (2012), which has been cited. Fresh smoke tends to have a lower lidar ratio at 532 nm.

. Line 203: "smallest sample size of the aerosol types" should be quantified.

The sample size of the aerosol types has been quantified in the supplemental document in Figure S3.

. Line 204: "This could indicate deficiencies in the model's optical properties for smoke, the transport, meaning the smoke plume is not in the correct location without the data assimilation, or uncertainties in the emissions" could be changed to "This could indicate deficiencies in the model's smoke optical properties and transport (i.e., the smoke plume is not in the correct location without the data assimilation), or uncertainties in the BB emissions."

This sentence has been modified as suggested.

. Line 207: "HSRL2 can have difficulty distinguishing between the two" This sentence needs more information and a reference.

A reference to Burton et al. (2012) has been added here.

. Line 211: "LARGE optical array is in situ and can provide a direct comparison between extinction and aerosol composition" this might be misleading if the AMS instrument is not operated by the LARGE group. Also, the extinction is for the total aerosol and composition is per species. The authors should clarify. The authors should also describe how these in situ measurements are selected i.e., airborne vertical profiles, constant altitude legs etc.

LARGE operated the AMS. All data points collected throughout the entire campaign (minus those with a cloud detected) were included.

. Line 213: "representative of fine particles that are efficiently sampled by the inlet"; "were subsampled such that only particles with an aerodynamic diameter less than 5 µm were included…"

This sentence has been modified as suggested.

. Line 234: "total aerosol mass concentration overestimated in GEOS". We recommend the authors show and evaluate the modeled total aerosol mass concentration profile.

The "total" aerosol mass concentration is not something that can be directly compared between the observations and model as this isn't an observed quantity. To avoid confusion, "total" has been removed from this sentence.

. Line 237: "an additional buildup of black carbon"

This has been fixed.

. Line 241: "This results in positive values for the analysis increment for black carbon mass" This sentence is not clear and should be re-written.

This sentence has been rephrased as "The assimilation of aerosol optical depth results in an increase of black carbon mass and…"

. Line 245: "Since brown carbon originates as a portion of what was organic carbon prior to GOCART2G, it is being included as organic carbon in the figure." This sentence is also not clear and should be re-written.

This sentence has been rephrased as "Since brown carbon was emitted as organic carbon prior to GOCART2G, it is being included as organic carbon in the figure."

. Line 247: "In general, there is not enough aerosol for these two species in the model." Could be replaced by "In general, these two aerosol species are underestimated in the model."

This sentence has been modified as suggested.

. Line 255: Figure 7d

Thanks for catching this!

. Line 265: Figure 9 seems to have minimal value in this paper and could be replaced by 1-2 sentences in the text. The relative humidity plot is not discussed, and the lowest altitude is not quantified on Fig. 9.

Figure 9 has been moved to the supplemental material.

. Line 268: "… optics look up tables are unchanged" -- This should be emphasized in Table 1 and in the description of Table 1.

This is now specified in the table caption.

. Line 269: "the aerosol mass concentration and relative humidity have the potential to differ in each of the model simulations" -- This should also clearly be stated in Table 1 and its description.

The local aerosol mass concentration and relative humidity can change by implementing a new convection scheme, which is indicated in the table.

. Line 270: "the relationship between the two and the optical properties remain the same." This sentence is not clear.

This sentence has been modified to "While the aerosol mass concentration and relative humidity have the potential to differ in each of the model simulations, the relationship between the two and the optical properties are the same among the model simulations."

. Line 274: "the ratio of sea salt" – the authors should consider replacing "ratio" by "fraction"

This has been updated as suggested.

. Line 277: "given the preference for coarse mode sea salt in GEOS (Bian et al., 2019)". This sentence is not clear.

This sentence has been modified to "However, given the bias toward excessive coarse mode sea salt in GEOS (Bian et al., 2019), we suspect this is not the case."

. Line 279: We recommend adding "in the model" after "The deficiency in sulphate and nitrate"

This has been updated as suggested.

. Table 2: Instead of "LARGE Observations", the authors should write the name of the instrument e.g., "Aerodyne HR-ToF-AMS"; and the four digit in "0.0677:1" do not seem necessary.

Observations presented in the table are from both PILS and the AMS, which is now noted in the caption. The number of significant digits for the ratio of black to organic carbon has been reduced.

. Line 286: "… are displayed in Figure 10"

This has been fixed.

. Line 287: "… which is always positive and representative of dry conditions"

This has been fixed.

. Line 289: "It is evident that GEOS needs a large bias in the mass concentration of organic carbon to accurately represent dry extinction." This sentence is not clear. The authors should rephrase. Also, they should consider quantifying the bias by providing an envelope around the 1:1 line and a percentage of points within this envelope.

RMSE and correlation have been added to each panel as well as the 2:1 line (black dashed lines).

[Figure]

. Line 310: The authors should provide the ranges of SSA values in Pistone et al. [2019]

Pistone et al. 2019 compares observed SSA from three instruments throughout the ORACLES field campaign, with the point made here that there is uncertainty in the observations as they do not necessarily agree with one another and that there is a spectral dependance on the SSA with higher values of SSA at smaller wavelengths. Giving the range of values reported by Pistone et al. 2019 would be counterproductive as the properties of the smoke (and large scale environment) are very different from the Philippines such that the observed values of SSA are not comparable between the two field campaigns.

. Line 312: "Nearly all observations have…"

This has been fixed as suggested.

. Line 324: "section 3.2"

Thanks for catching the typo!

. Line 326: The authors should describe the "chemical influence flag" and which gases it uses.

A sentence has been added to section 2.1 to describe the chemical influence flag.

. Line 330: The authors should explain why the size distribution is bi-modal for FIMS and unimodal per species in GEOS

There is a bi-modal distribution in the FIMS observations due to temporal and spatial variability in the particle size distribution. A figure has been added to the supplementary material that shows the time series of particle size distribution, altitude, RH, and the ratio of organic aerosol to sulfate. There are two regimes present in the timeseries shown: a smaller number of larger particles during lower altitude segments and a larger number of particles with a smaller radius during higher altitude segments. This variability cannot be represented by GOCART and is a limitation of the module.

[Figure]

GOCART has used the OPAC database since Chin et al. (2002), which was developed by Hess et al. (1998), and indicates a single log normal distribution for sulfate and carbon. Since the species are externally mixed, there is no interaction between the aerosol species that could cause variability in time or a bi-modal distribution.

. Line 357: "… parameterizations as well as…"

This has been fixed.

. Line 384: "there is evidence of this in the FIMS observations from CAMP2Ex" – this sentence needs more information.

This sentence has been modified to "Smoke is known to be internally mixed (Reid et al., 2006) and there is evidence of this in the FIMS observations from CAMP [2]Ex from variability in the particle size distribution."